# Parabrachial nucleus circuit governs neuropathic pain-like behavior

Li Sun [1,2,6 ✉], Rui Liu[1,2,6], Fang Guo[1,2], Man-qing Wen[1,2], Xiao-lin Ma[1,2], Kai-yuan Li[1,2], Hao Sun[3,4], Ceng-lin Xu [1], Yuan-yuan Li[1,2], Meng-yin Wu[5], Zheng-gang Zhu [1,2], Xin-jian Li[3], Yan-qin Yu[1,2], Zhong Chen [1], Xiang-yao Li [1,2] & Shumin Duan [1,2 ✉]

The lateral parabrachial nucleus (LPBN) is known to relay noxious information to the amygdala for processing affective responses. However, it is unclear whether the LPBN actively processes neuropathic pain characterized by persistent hyperalgesia with aversive emotional responses. Here we report that neuropathic pain-like hypersensitivity induced by common peroneal nerve (CPN) ligation increases nociceptive stimulation-induced responses in glutamatergic LPBN neurons. Optogenetic activation of GABAergic LPBN neurons does not affect basal nociception, but alleviates neuropathic pain-like behavior. Optogenetic activation of glutamatergic or inhibition of GABAergic LPBN neurons induces neuropathic pain-like behavior in naïve mice. Inhibition of glutamatergic LPBN neurons alleviates both basal nociception and neuropathic pain-like hypersensitivity. Repetitive pharmacogenetic activation of glutamatergic or GABAergic LPBN neurons respectively mimics or prevents the development of CPN ligation-induced neuropathic pain-like hypersensitivity. These findings indicate that a delicate balance between excitatory and inhibitory LPBN neuronal activity governs the development and maintenance of neuropathic pain.

[1] Department of Neurobiology and Department of Neurology of Second Affiliated Hospital, Zhejiang University School of Medicine, 310058 Hangzhou, China. [2] Research Units for Emotion and Emotion Disorders, NHC and CAMS Key Laboratory of Medical Neurobiology, MOE Frontier Science Center for Brain Research and Brain-Machine Integration, School of Brain Science and Brain Medicine, Zhejiang University, 310058 Hangzhou, China. [3] Department of Neurology of the Second Affiliated Hospital, Interdisciplinary Institute of Neuroscience and Technology, Zhejiang University School of Medicine, 310020 Hangzhou, China. [4] Key Laboratory of Biomedical Engineering of Ministry of Education, College of Biomedical Engineering and Instrument Science, Zhejiang University, 310027 Hangzhou, China. [5] Department of Epidemiology and Biostatistics, School of Public Health, Zhejiang University, 310058 Hangzhou, China. [6] These authors contributed equally: Li Sun, Rui Liu. ✉email: sunli2014@zju.edu.cn; duanshumin@zju.edu.cn

Neuropathic pain is a chronic condition caused by lesions of the peripheral or central nervous system, due to such factors as trauma, tumor invasion, stroke, diabetes, or multiple sclerosis[1,2]. The hallmarks of neuropathic pain include persistent hyperalgesia, allodynia, and affective responses due to sensitization of the somatosensory nervous system so that both noxious and innocuous stimuli are pathologically amplified[3,4]. Chronic pain can persist after the initial injury has healed and thus has been associated with plastic changes in the structure and function of the pain transmission pathway[5,6], particularly at the levels of primary afferents and the spinal cord[4,7–9]. In addition, the gate control theory of pain, which proposes that spinal nociceptive transmission neurons are gated by inhibitory inter-neurons in the dorsal horn[10], has also been applied to explain neuropathic pain. That is, peripheral injury may induce a dys-function of inhibitory dorsal horn neurons, leading to sensitiza-tion and increased excitability of projection neurons[11,12]. However, neuropathic pain also occurs in CNS injury or brain disorders[1,2], suggesting that higher brain regions may also play critical roles in neuropathic pain development. Indeed, the supraspinal and cortical areas including the primary somatosen-sory cortex and the anterior cingulate cortex have been also reported to be involved in the neuropathic pain[3,4,7–9,13].

Two ascending pathways have been described for pain trans-mission to higher centers. The spinothalamic tract carries sensory inputs from spinal neurons to the thalamus, which relays the information to the somatosensory cortex for processing the sen-sory and discriminative aspects of pain, whereas the spinopar-abrachial tract transmits the pain signal to the parabrachial nucleus (PBN)[7,14–19], which is reported to project to the amygdala and other brain regions to transform the sensory signals into the aversive and emotional components of pain[20–29]. Furthermore, the PBN has been associated with other important sensory processes, such as itch and craniofacial affective pain, as well as aversive emotional behaviors, including aversive learning, avoidance behavior, and anorexia/hunger[30–36]. However, the role of PBN in the development and maintenance of neuropathic pain, a persis-tent pain characterized by prominent emotional responses[11,37,38], is not clear.

Using common peroneal nerve (CPN) ligation[39] as a neuro-pathic pain model combined with $Ca^{2+}$ signal imaging[40], an optogenetics approach[41], and behavioral testing[20,42,43], we found that glutamatergic neurons in the lateral PBN (LPBN) are responsible not only for relaying basal nociception, but also for the processing of neuropathic pain. On the other hand, GABAergic neurons, which constitute only 10.1% of LPBN neurons, are not involved in the transmission of basal nociception signals. However, the activity of GABAergic LPBN neurons plays a gating role in the development and transmission of neuropathic pain-like hypersensitivity. Trans-synaptic virus tracing and elec-trophysiological studies identified direct synaptic innervation of and functional inhibition on glutamatergic LPBN neurons by local GABAergic neurons. These results, together with the finding that CPN ligation selectively activated glutamatergic LPBN neu-rons indicate that GABAergic LPBN neurons gate the sensitiza-tion of glutamatergic neurons, which is necessary and sufficient for the development and transmission of neuropathic pain.

## Results

### Increased activity of glutamatergic LPBN neurons in neuro-pathic pain-like hypersensitivity. Successful induction of neu-ropathic pain-like hypersensitivity was confirmed seven days after CPN ligation by a marked reduction in the paw withdrawal threshold (PWT) upon stimulation of the lateral aspect of the plantar paw surface by a von Frey filament (Supplementary

Fig. 1a–c). Immunostaining of c-Fos one week after CPN ligation showed a dramatic increase in the density of c-Fos-positive cells co-expressing $Ca^{2+}$/calmodulin-dependent protein kinase IIα (CaMKIIα, a marker of glutamatergic neurons) in the LPBN as compared with Sham operation, whereas no significant difference was found between CPN-treated and Sham-treated mice in c-Fos-positive cells co-expressing glutamic acid decarboxylase 67 (GAD67, a marker of GABAergic neurons) (Fig. 1a–d). Thus, after CPN ligation, ~41.8% of CaMKIIα neurons were activated (co-localized with c-Fos) as compared with ~10.3% in Sham-treated mice, and only ~8.5% of $GAD67^+$ neurons co-expressed c-Fos as compared with ~6.3% in Sham-treated mice (Fig. 1d).

We targeted glutamatergic neurons expressing vesicular gluta-mate transporter 2 (VgluT2) by local injection of adeno-associated viruses (AAVs) containing a Cre-dependent eYFP tag into the LPBN of a mouse line expressing Cre recombinase under the control of the VgluT2 promoter (VgluT2-ires-Cre). We found that ~83% of CaMKIIα-positive neurons in the superior LPBN (PBsl) overlapped with VgluT2 and ~82.1% of CaMKIIα-positive neurons in the dorsal LPBN (PBdl) overlapped with VgluT2. On the other hand, ~78% of $VgluT2^+$ neurons in the PBsl overlapped with CaMKIIα and ~76.8% of $VgluT2^+$ neurons in the PBdl overlapped with CaMKIIα (Fig. 1e–f), suggesting that $CaMKIIα^+$ and $VgluT2^+$ neurons overlap extensively in the LPBN. To estimate the ratio of GABAergic/glutamatergic neurons in the LPBN, we applied fluorescence in situ hybridization using GAD1 and Slc17a6 (VgluT2) probes (Fig. 1g). We found that the ratio of GABAergic neurons (GAD1-positive cells/GAD1-positive cells + VgluT2-posi-tive cells) was only 10.1% (Fig. 1h), consistent with the data shown in the Allen Brain Atlas (http://www.brain-map.org).

The above results indicate that glutamatergic, but not GABAergic neurons in the LPBN are activated during the development of neuropathic pain-like hypersensitivity. To investigate changes in the dynamic activity of LPBN neurons associated with neuropathic pain-like hypersensitivity, we used in vivo fiber photometry[40] to monitor their $Ca^{2+}$ levels. We thus injected AAVs encoding Cre-dependent driven eYFP or GCaMP7s, a genetically-encoded fluorescent $Ca^{2+}$ indicator, into the LPBN of VgluT2-ires-Cre (Fig. 1i) or GAD2-ires-Cre mice (Fig. 1l). Three weeks after virus injection, mice were subjected to CPN ligation or Sham operation. We found that a noxious pinch induced a small but significant $Ca^{2+}$ elevation in both glutamatergic (Fig. 1j, k) and GABAergic neurons (Fig. 1m, n) in the LPBN in Sham-operated mice. However, in CPN-ligated mice the same stimulus induced dramatic $Ca^{2+}$ elevation in glutamatergic (Fig. 1j, k) but not in GABAergic neurons (Fig. 1m, n), confirming that glutamatergic LPBN neurons are selectively activated by the noxious stimulus.

To further characterize the $Ca^{2+}$ signals in glutamatergic LPBN neurons associated with neuropathic pain-like hypersensi-tivity, we used in vivo miniaturized microscopy, which has high sensitivity and spatial resolution so that weak $Ca^{2+}$ responses in individual neurons can be detected[44,45]. Three weeks after virus injection, activity of glutamatergic LPBN neurons in intact freely-moving mice was assessed with pinch or von Frey stimulation (pre-CPN) (Supplementary Fig. 2a–c). Subsequently, VgluT2-ires-Cre mice were subjected to CPN surgery. We found that a noxious stimulus (pinch) and a strong mechanical stimulus (1.0 g von Frey) of the hindpaw elicited detectable $Ca^{2+}$ responses in control mice (pre-CPN) and such responses were significantly higher one week after CPN surgery (post-CPN) than pre-CPN in the same mice (Supplementary Fig. 2d, e). Interestingly, a weak mechanical stimulus (0.4 g von Frey) induced an apparent $Ca^{2+}$ response in post-CPN but not in pre-CPN mice (Supplementary Fig. 2f, g), consistent with the mechanical allodynia of neuropathic pain.

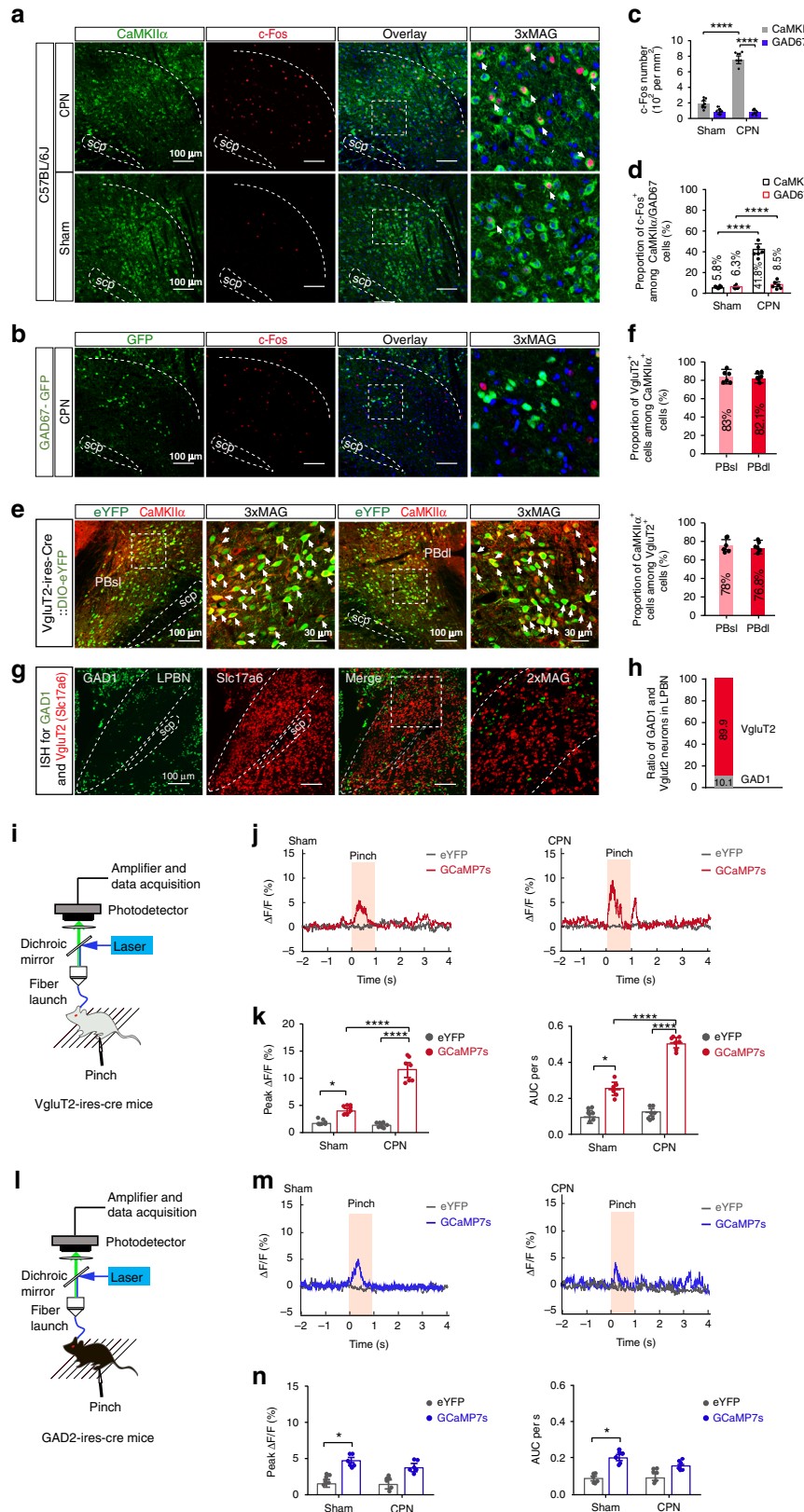

**Activation of glutamatergic LPBN neurons is sufficient to induce allodynia, hyperalgesia, and the place-avoidance response**. Because glutamatergic LPBN neurons were selectively activated in neuropathic pain-like hypersensitivity, we next determined whether the direct optogenetic activation of these neurons is sufficient to induce allodynia and hyperalgesia,

hallmarks of neuropathic pain. We targeted the glutamatergic neurons by injection of AAVs delivering a construct that contained Cre-dependent channelrhodopsin-2 (ChR2) coupled to an eYFP tag into the LPBN of VgluT2-ires-Cre mice (Fig. 2a, c). Mice injected with AAVs containing only the eYFP tag served as controls. Current-clamp recordings from brain slices showed that blue

**Fig. 1 CPN ligation activates glutamatergic LPBN neurons. a** Representative images of co-labeling (arrowheads) of CaMKIIα-positive neurons and c-Fos immunoreactivity in the LPBN (white dashed lines). Rightmost panels, 3 times magnification of the boxed areas in the left panels. Scale bars, 100 μm. **b** Example images showing c-Fos positive (red) and GAD67-GFP-expressing cells in the LPBN one week after CPN ligation. Rightmost panel, 3 times magnification of the boxed area in the left panel. Blue, DAPI staining. Scale bars, 100 μm. **c** Density of c-Fos-positive cells in CaMKIIα-expressing and GAD67-GFP-expressing cells in the LPBN. **d** Proportions of CaMKIIα or GAD67-GFP-positive cells co-expressed with c-Fos in the LPBN. **e** Representative images of neurons expressing VgluT2-eYFP co-localize (arrows) with the CaMKIIα in PBsl (left two panels) and PBdl (right two panels). 3xMAG (the second and fourth panels), 3 times magnification of the boxed area in the first and the third panels, respectively. Scale bars, 100 μm and 30 μm (3xMAG). **f** Percentage of CaMKIIα-positive neurons that co-express VgluT2 (upper) and percentage of VgluT2-positive neurons that co-express CaMKIIα (lower). $n = 898$ (PBsl) and 856 (PBdl) neurons from 5 sections from 5 mice. **g** Representative in situ hybridization for the GAD1 (green) and VgluT2 (red, Slc17a6) in LPBN. Scale bars, 100 μm. Rightmost panels, 2× magnification of the boxed area in the left panel. **h** Ratio of Vglut2 or GAD1-positive neurons in the LPBN as in **g**. $n = 6$ sections from 6 mice. **i, l** Schematic of the recording system for the $Ca^{2+}$ signal in LPBN neurons with fiber photometry in VgluT2-ires-Cre mice (**i**) and GAD2-ires-Cre mice (**l**). **j, m** Representative fluorescence signals ($\Delta F/F$) of GCaMP7s (red) and eYFP (dark gray) recorded from glutamatergic (**j**) or GABAergic (**m**) LPBN neurons transfected with DIO-GCaMP7s or DIO-eYFP aligned to the pinch stimulation. **k, n** Averaged peak $\Delta F/F$ (left) and area under the curve (AUC) per second (right) of GCaMP7s and eYFP fluorescence signals from glutamatergic (**k**) or GABAergic (**n**) LPBN neurons. All data are presented as mean ± s.e.m. and error bars represent s.e.m. *$P < 0.05$ and ****$P < 0.0001$. See also Supplementary Table 1 for further statistical information. Source data are provided as a Source Data file.

laser pulses induced time-locked action potential firing in ChR2-eYFP-infected neurons in the LPBN (LPBN$^{VgluT2}$) (Fig. 2b), establishing the expression and function of ChR2 in these neurons. Consequently, a dramatic increase in c-Fos expression was found in LPBN after light stimulation of the LPBN$^{VgluT2}$ neurons (Supplementary Fig. 3a, b). The Von Frey and Hargreaves tests were used to assess the mechanical threshold and thermal response latency of the paw-withdrawal response[43,46]. We found that the optical activation of LPBN$^{VgluT2}$ neurons expressing ChR2-eYFP, but not those expressing eYFP, with blue laser light (473 nm, 20 Hz, 5 mW) induced marked mechanical allodynia, as manifested by a drastic decrease in the threshold for the paw-withdrawal response induced by von Frey filaments in Sham-operated mice (Fig. 2d). Moreover, photoactivation of LPBN$^{VgluT2}$ neurons expressing ChR2-eYFP, but not those expressing eYFP, also induced thermal hyperalgesia in Sham-operated mice, as manifested by a significant decrease in the latency of the paw-withdraw response to thermal stimulation in the Hargreaves test (Fig. 2e). Similarly, illumination of LPBN$^{CaMKIIα}$ neurons expressing ChR2-mCherrry, but not those expressing mCherry, also induced both mechanical allodynia and thermal hyperalgesia in Sham-operated mice (Fig. 3a–c). Interestingly, this light-induced mechanical and thermal hyperalgesia was time-locked to the light stimulation, since both the mechanical threshold and thermal latency of paw withdrawal responses returned to the control level after the stimulation (Fig. 3b, c). The light-induced activation of LPBN$^{CaMKIIα}$ neurons was confirmed by assessing the c-Fos expression level in the LPBN (Supplementary Fig. 3c, d).

Next, we used the real-time place avoidance (RTPA) assay[47,48] to determine whether glutamatergic LPBN neurons are involved in the affective component of pain behavior. Mice were habituated and placed in a two-chamber arena. The time spent in free exploration in each chamber was recorded for 10 min (baseline), followed by a 10-min recording during conditioned photostimulation (20 Hz, 5-ms pulse width, 5 mV) of LPBN$^{VgluT2}$ or LPBN$^{CaMKIIα}$ neurons when the mouse was in its preferred chamber, and followed again by a 10-min recording after photostimulation (Fig. 2f). As expected, upon optogenetic activation of the LPBN$^{VgluT2}$ or LPBN$^{CaMKIIα}$ neurons, mice tended to flee to the opposite chamber, and subsequently they moved less and spent significantly more time in the unstimulated chamber, a behavior that persisted even during the 10-min post-photostimulation period (Fig. 2i, j and Fig. 3e), whereas illumination had no effect on the RTPA test in control DIO–eYFP or CaMKIIα–mCherry mice (Fig. 2g, h and Fig. 3d). On the other

hand, 5 min optogenetic activation of LPBN$^{CaMKIIα}$ neurons did not affect the total distance moved and the ratio of time spent in the periphery and center of the open field in naïve wild-type mice (Fig. 3f–h), indicating that the locomotor and baseline anxiety levels were not altered by optogenetic manipulation. Altogether, these data demonstrated that the optogenetic activation of glutamatergic LPBN neurons is sufficient to induce both hyperalgesia and affective aversion in naïve mice, mimicking the typical symptoms of neuropathic pain. Although chronic pain and anxiety are often comorbid, probably due to long-term plasticity changes in some brain structures[9,49], our results indicate that short-term stimulation of LPBN neurons is not sufficient to induce anxiety behavior.

**Inactivation of glutamatergic LPBN neurons alleviates both basal nociception and neuropathic pain-like hypersensitivity.** To further illustrate the correlation between glutamatergic neuronal activity in the LPBN and neuropathic pain-like hypersensitivity, we injected AAV–CaMKIIα–eNpHR3.0–mCherry virus bilaterally into the LPBN of wild-type mice or AAV-DIO-eNpHR-eYFP virus into the LPBN of VgluT2-ires-Cre mice to selectively silence the activity of LPBN$^{CaMKIIα}$ or LPBN$^{VgluT2}$ neurons, respectively, during illumination by a 589-nm laser (Figs. 4a, 5a). We found that the eNpHR-mediated photosilencing of LPBN$^{CaMKIIα}$ or LPBN$^{VgluT2}$ neurons drastically elevated the PWT in the von Frey test in both Sham-operated and CPN-ligated mice (Figs. 4b, 5b). An increase in paw-withdraw latency in response to thermal stimulation in the Hargreaves test also occurred in Sham-operated and CPN-ligated mice upon photo-silencing of LPBN$^{CaMKIIα}$ or LPBN$^{VgluT2}$ neurons (Figs. 4c, 5c). These results indicate that glutamatergic LPBN neurons mediate both basal nociception and neuropathic pain-like hypersensitivity.

To estimate the aversive quality of ongoing or tonic pain, we next tested real-time place preference (RTPP), which is associated with the seeking of pain relief[50,51,52]. We used this assay to determine whether photosilencing of LPBN$^{CaMKIIα}$ or LPBN$^{VgluT2}$ neurons after CPN ligation elicits pain-relief-seeking behavior. Each mouse was first habituated to the two connected chambers for 10 min, paired with light stimulation in one chamber and no light in the opposite chamber for another 10 min, and again no light stimulation for 10 min (Fig. 4d). Strikingly, we found that photoinhibition of the LPBN$^{CaMKIIα}$ or LPBN$^{VgluT2}$ neurons significantly increased the time spent in the chamber paired with light stimulation during the light and post-light stimulation periods in CPN-ligated mice but not in Sham-operated mice (Fig. 4e, f, g and Fig. 5d–f). Importantly, we

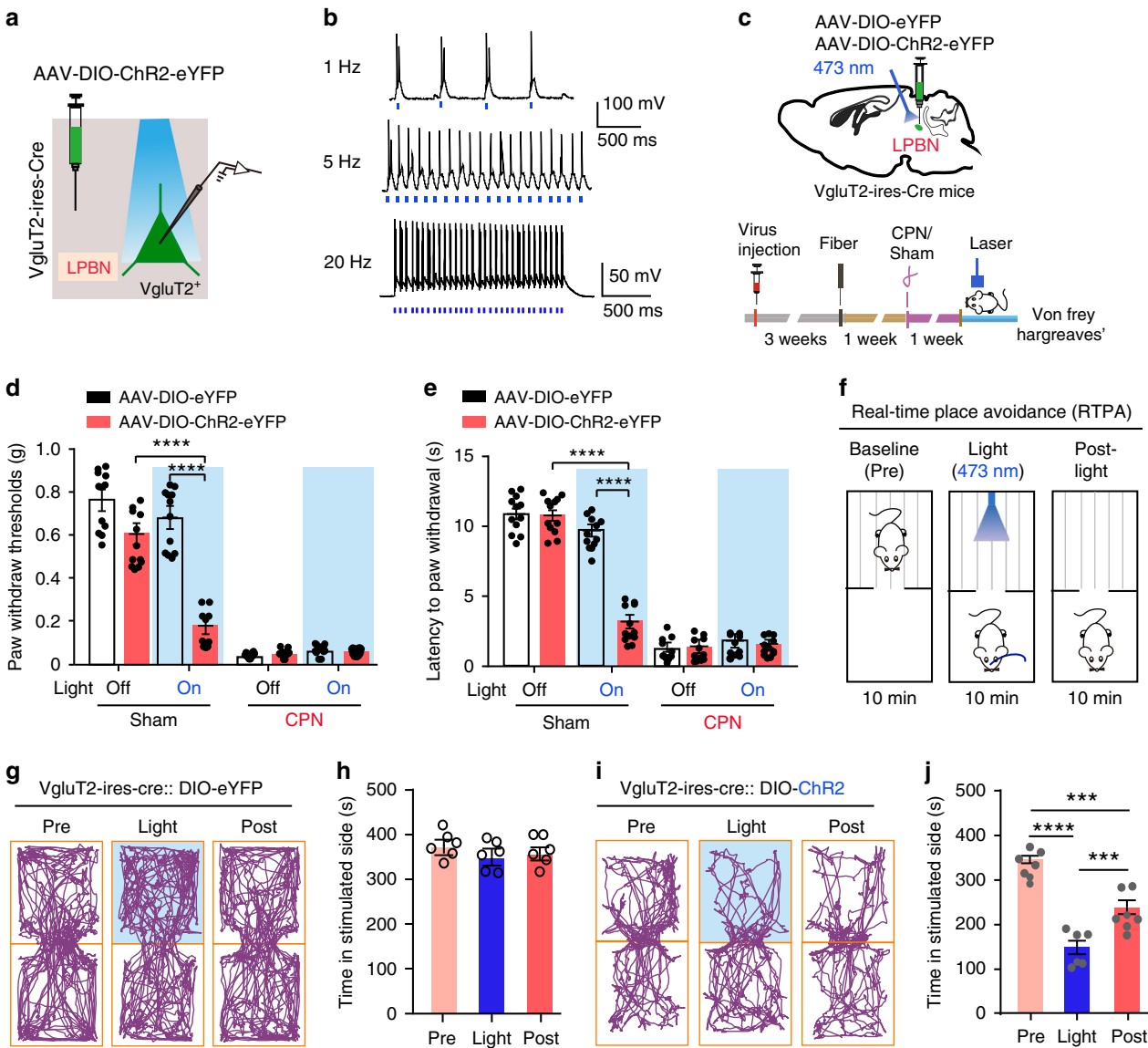

**Fig. 2 Optogenetic activation of VgluT2 neurons in LPBN induces hyperalgesia and place avoidance behavior. a** Schematic of light stimulation of and patch-clamp recording from glutamatergic LPBN neurons transfected with AAV-Cre-on-ChR2-eYFP in brain slices from VgluT2-ires-Cre mice. **b** Example patch-clamp recordings showing light stimulation-induced time-locked action potential firing in ChR2-expressing VgluT2 neurons in the LPBN. Blue bars indicate application of light stimuli (473 nm, 5 ms, ~5 mW) at 1 Hz (upper), 5 Hz (middle), and 20 Hz (lower). **c** Schematic of stereotaxic delivery of AAV carrying Cre-dependent ChR2 into the LPBN of VgluT2-ires-Cre mice (upper) and experimental design and timeline of the behavioral experiments (lower). **d** Illumination (473 nm, 20 Hz, 5-ms pulse width, ~5 mW) of the LPBN significantly decreases the paw-withdrawal thresholds (PWT) in response to von Frey mechanical stimulation in Sham-operated mice transfected with ChR2-eYFP, but not in mice transfected with eYFP in VgluT2 LPBN neurons. **e** Illumination (473 nm, 20 Hz, 5-ms pulse width, ~5 mW) of the LPBN significantly decreases the latency of the thermal paw-withdrawal response in the Hargreaves test in Sham-operated mice transfected with ChR2-eYFP, but not in mice transfected with eYFP in VgluT2 LPBN neurons. **f** Schematic of the real-time place avoidance (RTPA) test. **g–j** Representative tracks (**g, i**) and quantification of time spent in the preferred chamber (**h, j**) in the RTPA test before (Pre), during (Light), and immediately after (Post) laser illumination (473 nm, 20 Hz, 5-ms pulse width, ~5 mW) of the LPBN transfected with eYFP (**g, h**) or ChR2 (**i, j**) in Vgltu2-ires-Cre mice. All data are presented as mean ± s.e.m. and error bars represent s.e.m. ***P < 0.001 and ****P < 0.0001. See also Supplementary Table 1 for further statistical information. Source data are provided as a Source Data file.

found that photoinhibition of LPBN$^{CaMKII\alpha}$ or LPBN$^{VgluT2}$ neurons did not affect the total distance moved and the ratio of time spent in the periphery and center of the open field in Sham-operated or CPN-ligated mice, suggesting that the locomotor and anxiety levels were not altered by optogenetic manipulation (Figs. 4h–m, 5g–k). Taken together, these data demonstrate that silencing LPBN$^{CaMKII\alpha}$ or LPBN$^{VgluT2}$ neurons not only alleviates basal nociception but also relieves both the sensory and affective components of neuropathic pain-like hypersensitivity.

**Activation of GABAergic LPBN neurons alleviates neuropathic pain-like hypersensitivity but not basal nociception.** The results that GABAergic neurons constitute only 10.1% of LPBN neurons and that most of the neurons activated in the LPBN after CPN ligation are glutamatergic (Fig. 1a–h) prompted us to investigate whether the activation of local GABAergic neurons has any effect on basal nociception and neuropathic pain-like hypersensitivity. To address this issue, optical fibers were implanted bilaterally into the LPBN of transgenic mice with vesicular GABA transporter (vGAT)-ChR2 (H134R)-eYFP (vGAT-ChR2). The activation of

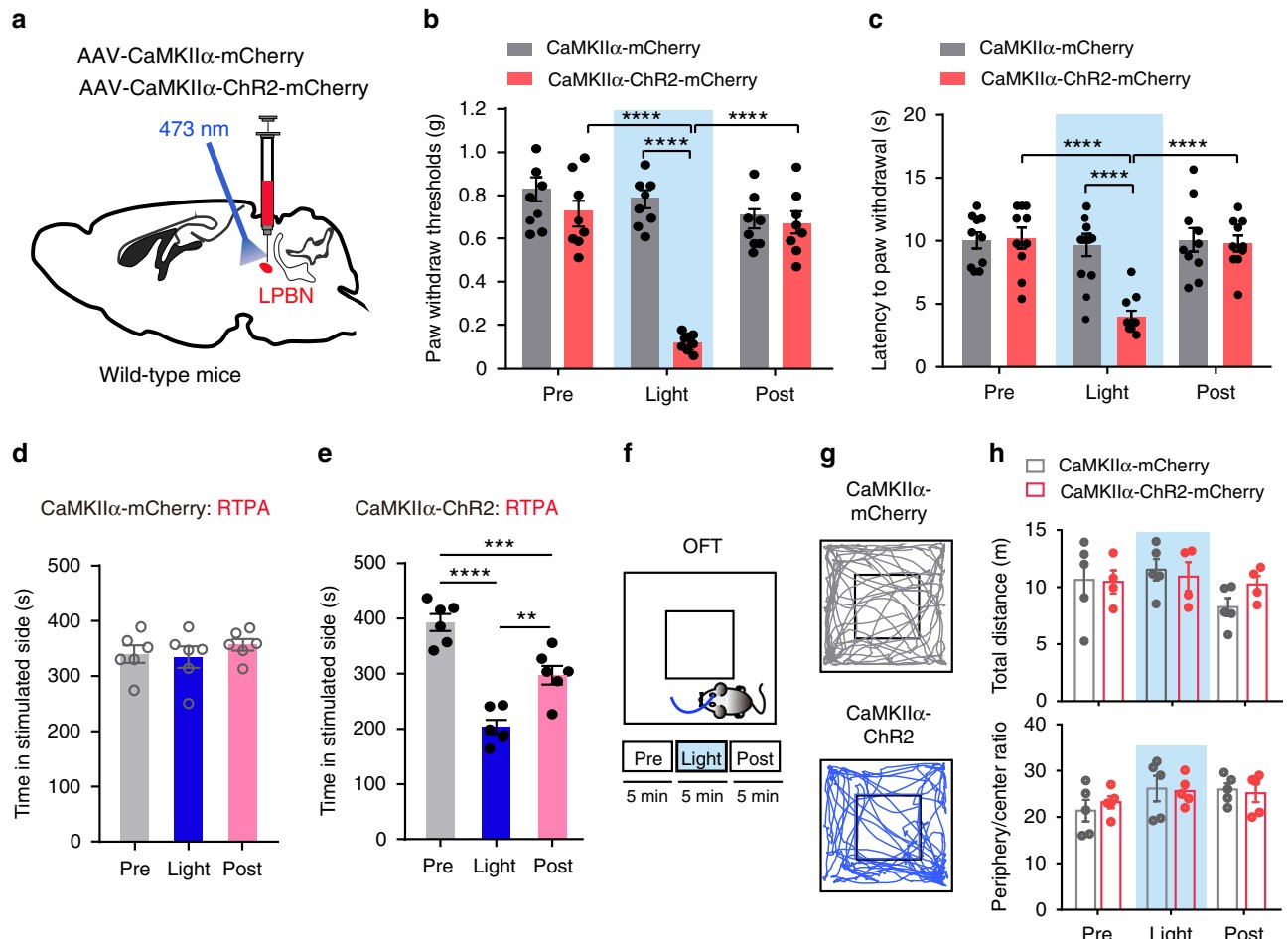

**Fig. 3 Optogenetic activation of LPBN$^{CaMKII\alpha}$ neurons induces hyperalgesia and place avoidance behavior. a** Schematic of the stereotaxic delivery of AAV carrying CaMKIIα-ChR2 into the LPBN of wild-type mice. **b** Light activation (473 nm, 20 Hz, 5-ms pulse width, ~5 mW) of LPBN$^{CaMKII\alpha}$ neurons significantly decreases the PWT induced by von Frey stimulation in mice injected with AAV-CaMKIIα-ChR2-mCherry, compared with mice injected with AAV-CaMKIIα-mCherry. No significant difference was found between before (Pre) and after (Post) light stimulation. **c** Light activation (473 nm, 20 Hz, 5-ms pulse width, ~5 mW) of LPBN$^{CaMKII\alpha}$ neurons significantly decreases thermal paw-withdrawal latency measured by the Hargreaves test in mice injected with AAV-CaMKIIα-ChR2-mCherry, compared with mice injected with AAV-CaMKIIα-mCherry. No significant difference was found between before (Pre) and after (Post) light stimulation. **d, e** Quantification of time spent in the preferred chamber in the RTPA test before (Pre), during (Light), and after (Post) 10-min illumination of the LPBN in mice transfected with AAV-CaMKIIα-mCherry (**d**) and AAV-CaMKIIα-ChR2-mCherry (**e**). **f** Schematic of the open field test (OFT) with photostimulation via a 473-nm laser (Pre, Light, and Post; 5 min for each stage). **g** Example traces of the OFT from mice with AAV-CaMKIIα-mCherry (upper) and AAV-CaMKIIα-ChR2-mChreey (lower) injected into the LPBN. **h** Quantification of the total distance moved (upper) and ratio of time spent in the periphery and center (lower) in the OFT before (Pre), during (Light), and after (Post) 10-min illumination of the LPBN in wild-type mice transfected with AAV-CaMKIIα-mCherry and AAV-CaMKIIα-ChR2-mCherry. All data are presented as mean ± s.e.m. and error bars represent s.e.m. ***$P < 0.001$ and ****$P < 0.0001$. See also Supplementary Table 1 for further statistical information. Source data are provided as a Source Data file.

GABAergic neurons by photostimulation in the LPBN was confirmed by double-labeling for vGAT and c-Fos (Fig. 6a). Surprisingly, we found that although optogenetic activation of GABAergic LPBN neurons failed to affect the basal nociception in Sham-operated mice, it induced a significant, time-locked increase in the threshold and latency of the paw-withdrawal responses evoked by mechanical and thermal stimulation in CPN-ligated mice to levels comparable to those in Sham-operated mice (Fig. 6b, c). Furthermore, light stimulation of the GABAergic LPBN neurons significantly increased the time spent in the chamber paired with photostimulation in CPN-ligated mice, but not in controls, in the RTPP test (Fig. 6d–g), consistent with the light stimulation-induced pain relief in CPN-ligated mice.

Since the LPBN also receives GABAergic innervation from other brain regions[28,53,54], it was possible that light stimulation in the vGAT-ChR2 transgenic mice activated the fibers and

terminals of GABAergic neurons projecting to or via the LPBN from other brain regions. To determine whether activation of the local GABAergic LPBN neurons is sufficient to inhibit neuropathic pain-like hypersensitivity, we initially injected Cre-dependent AAV-DIO-ChR2-eYFP or AAV-DIO-eYFP virus into the LPBN of mice expressing Cre recombinase under the control of the glutamic acid decarboxylase enzyme (GAD2-ires-Cre mice) (Fig. 6h and Supplementary Fig. 4). A dramatic increase in the threshold and latency of the paw-withdrawal responses induced by mechanical and thermal stimulation was found in CPN-ligated mice during illumination of LPBN neurons transfected with AAV-DIO-ChR2-eYFP in GAD2-ires-Cre mice, but neither in CPN-ligated mice transfected with AAV-DIO-eYFP nor in Sham-operated mice (Fig. 6i, j). Similarly, the time spent in the chamber paired with photostimulation was significantly increased in CPN-ligated mice in the RTPP test during and 10 min after light stimulation of the LPBN transfected with AAV-DIO-ChR2-eYFP,

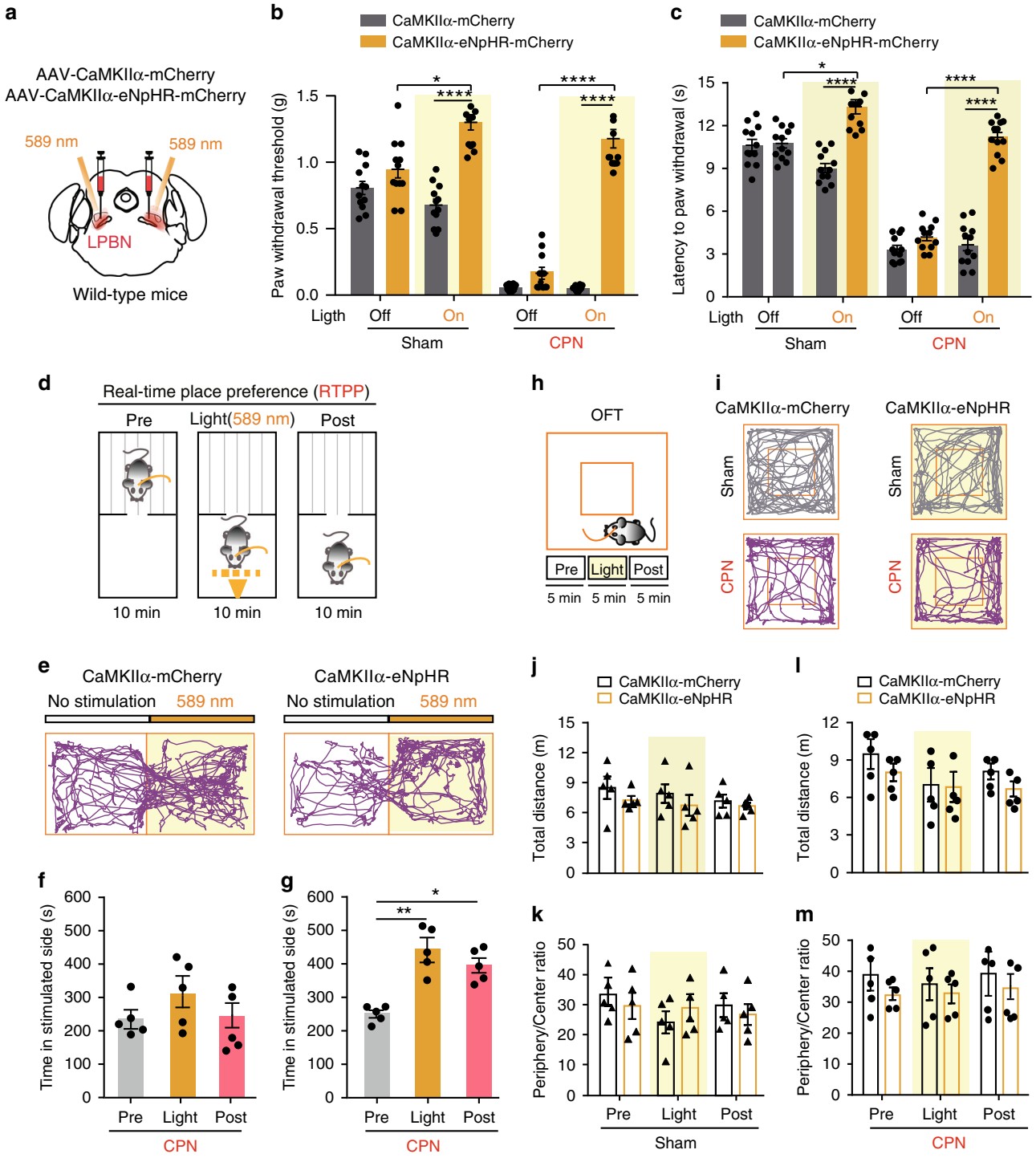

but not with AAV-DIO-eYFP (Fig. 6k). Optogenetic activation of GABAergic LPBN neurons in CPN-ligated mice did not affect the total distance moved and the ratio of time spent in the periphery and center of the open field (Fig. 6l), suggesting that the locomotor and anxiety levels were not altered by optogenetic manipulation. These results demonstrated that activation of local GABAergic LPBN neurons, while leaving the basal nociception unaffected, efficiently alleviates neuropathic pain-like hypersensitivity in terms of both sensory discrimination and emotional behavior.

We next used the pharmacogenetic approach to further confirm the effect of activating GABAergic LPBN neurons on neuropathic pain-like hypersensitivity. We bilaterally injected the LPBN of GAD2-ires-Cre mice with an AAV expressing hM3Dq, a designer receptor exclusively activated by designer drugs (DREADD)[55] or mCherry as control (Fig. 6m). Three weeks after virus infection, CPN ligation was carried out, and a week later, the hM3Dq agonist clozapine-N-oxide (CNO) was injected intraperitoneally (i.p.) once per day for 4 days (Fig. 6m). As expected, the decreased PWT induced by CPN ligation was elevated on the first day of CNO injection and was further elevated after the subsequent daily injections in mice expressing hM3Dq in the GABAergic LPBN neurons, but not in mice transfected with mCherry (Fig. 6n). Interestingly, delivery of picrotoxin (PTX, a blocker of the GABA$_A$ receptor chloride channel) into the LPBN via an intracerebral cannula completely

**Fig. 4 Photoinhibition of LPBN$^{CaMKII\alpha}$ neurons alleviates both basal nociception and neuropathic pain-like hypersensitivity. a** Schematic of the light illumination and virus injection strategy for the expression of CaMKIIα-eNpHR or CaMKIIα-mCherry in the LPBN of wild-type mice. **b** Yellow illumination (589 nm, 1 Hz, 999-ms pulse width, 10 mW) of the LPBN significantly elevates the PWT in response to mechanical stimulation in both Sham-operated and CPN-ligated mice transfected with AAV-CaMKIIα-eNpHR, but not in mice transfected with CaMKIIα-mCherry in the LPBN. **c** Yellow illumination (589 nm, 1 Hz, 999-ms pulse width, 10 mW) of the LPBN significantly increases the latency of the thermal paw-withdrawal response in the Hargreaves test in both Sham-operated and CPN-ligated mice transfected with AAV-CaMKIIα-eNpHR, but not in mice transfected with CaMKIIα-mCherry in the LPBN. **d** Schematic of the real-time place preference (RTPP) test. **e** Representative tracks of RTPP illustrating yellow (589 nm) light-evoked behavioral preference in CPN-ligated mice injected with AAV-CaMKIIα-mCherry (left) or CaMKIIα-eNpHR (right) virus in the LPBN. **f, g** Quantification of RTPP in CPN-ligated mice before (Pre), during (Light), and after (Post) 10-min yellow illumination of the LPBN in mice injected with AAV-CaMKIIα-mCherry (**f**) or CaMKIIα-eNpHR (**g**). **h** Schematic of the open field test (OFT) before (Pre), during (Light), and after (Post) 5 min photoinhibition (589 nm, 1 Hz, 999-ms pulse width, 10 mW) of LPBN$^{CaMKII\alpha}$ neurons. **i** Representative tracks of mice transfected with CaMKIIα-mCherry (left) or CaMKIIα-eNpHR (right) in the LPBN with yellow laser illumination (589 nm) in Sham-operated (upper) or CPN-ligated (lower) wild-type mice. **j–m** Quantification of total distance moved (**j, l**) and ratio of time spent in the periphery and center (**k, m**) in the OFT before (Pre), during (Light), and after (Post) 5 min yellow light illumination of the LPBN in mice transfected with CaMKIIα-mCherry or CaMKIIα-eNpHR in Sham-operated (**j, k**) and CPN-ligated (**l, m**) mice. All data are presented as mean ± s.e.m. and error bars represent s.e.m. *$P < 0.05$, **$P < 0.01$, and ****$P < 0.0001$. See also Supplementary Table 1 for further statistical information. Source data are provided as a Source Data file.

reversed the effect of CNO (Fig. 6n), indicating that GABAergic neurons control neuropathic pain-like hypersensitivity through locally-released GABA in the LPBN, rather than through their projections to other regions outside the LPBN. We found that pharmacogenetic activation of GABAergic LPBN neurons by CNO in CPN-operated mice also induced a conditioned place preference (CPP) in the chamber paired with i.p. injection of CNO (Supplementary Fig. 5a–c), indicating the relief of the aversive emotion accompanying neuropathic pain-like hypersensitivity. Activation of GABAergic LPBN neurons by CNO treatment was confirmed by immunostaining of c-Fos expression in the LPBN (Supplementary Fig. 5d, e).

**Inactivation of GABAergic LPBN neurons is sufficient to induce hyperalgesia and the place avoidance response.** The above results—that activation of GABAergic LPBN neurons alleviates neuropathic pain-like hypersensitivity through local inhibition of glutamatergic neuronal activity and that activation of glutamatergic LPBN neurons is sufficient to induce hyperalgesia mimicking neuropathic pain—prompted us to investigate whether the inactivation of GABAergic LPBN neurons affects pain processing. To do this, AAVs carrying Cre-dependent GtACR1 (Guillardia theta anion channel rhodopsins 1)[56] was injected into the LPBN of GAD2-ires-Cre mice (Fig. 7a). In contrast to the activation of GABAergic LPBN neurons, which did not affect the basic pain threshold in Sham-operated mice (Fig. 6b, c), we found that light inactivation of GABAergic LPBN neurons drastically decreased the threshold and latency of the paw-withdrawal responses evoked by mechanical allodynia (Fig. 7b) and thermal stimulation (Fig. 7c) in naïve (without CPN ligation) GAD2-ires-Cre mice transfected with GtACR1-eYFP to a level comparable to that in CPN-ligated mice, while illumination of GABAergic LPBN neurons transfected with eYFP had no effect (Fig. 7b, c). Furthermore, illumination of GABAergic LPBN neurons transfected with GtACR1-eYFP, but not with eYFP, significantly decreased the time spent in the chamber paired with illumination in the RTPA test (Fig. 7d, e). Optogenetic inactivation of GABAergic LPBN neurons did not affect the total distance moved and the ratio of time spent in the periphery and center in the open field test (Fig. 7f), indicating that the motor performance and baseline anxiety level were not changed by optogenetic manipulation.

Consistent with this, we found that light inactivation of GABAergic LPBN neurons transfected with GtACR1-eYFP induced a robust increase (~4.0-fold) in c-Fos expression in glutamatergic (CaMKIIα$^+$) but not in GABAergic LPBN neurons, compared with mice transfected with eYFP (Fig. 7g–i). Similar to the CPN-ligation-induced c-Fos expression (Fig. 1a–d), ~80.1% of

the c-Fos-positive neurons overlapped with CaMKIIα, while only ~2.6% of the c-Fos-positive neurons overlapped with eYFP (GABAergic neurons) (Fig. 7j). These results demonstrate that, similar to optogenetic activation of the glutamatergic LPBN neurons (Fig. 2d, e), optogenetic inactivation of the GABAergic LPBN neurons also mimics the typical symptoms of neuropathic pain, suggesting that tonic activity of GABAergic neurons is critical for preventing neuropathic pain-like hypersensitivity.

**Monosynaptic innervation of glutamatergic LPBN neurons by local GABAergic neurons.** The above results suggest that GABAergic LPBN neurons modulate neuropathic pain-like hypersensitivity by directly inhibiting glutamatergic LPBN neurons. To address this possibility, we first used a monosynaptic retrograde tracing viral tool by injecting Cre-dependent expression helper virus including AAV-CAG-DIO-TVA-eGFP and AAV-CAG-DIO-RG on day 1 and RV-EnvA-DsRed on day 21 into the LPBN of VgluT2-ires-Cre mice (Fig. 8a). We found that some trans-synaptically infected LPBN neurons expressing DsRed were GABAergic neurons as identified by in situ hybridization for GAD1 mRNA expression (Fig. 8b), suggesting that glutamatergic LPBN neurons receive direct monosynaptic innervation from local GABAergic neurons.

To further electrophysiologically demonstrate the direct synaptic innervation of LPBN$^{CaMKII\alpha}$ neurons by local GABAergic interneurons, we simultaneously injected two types of virus, AAV-CaMKIIα-mCherry and AAV-DIO-ChR2-eYFP, into the LPBN of GAD2-ires-Cre mice (Fig. 8c, d). Voltage-clamp recordings from LPBN$^{CaMKII\alpha}$ neurons expressing mCherry in brain slices showed that light stimulation of GABAergic neurons expressing ChR2-eYFP optically evoked inhibitory postsynaptic currents (eIPSCs), which were evident as an inward current when neurons were recorded with a high Cl$^-$ intra-pipette solution (135 mM), at a holding membrane potential of –70 mV (Fig. 8e). Several lines of evidence indicated that this light-evoked synaptic current resulted from the direct activation of monosynaptically-connected GABAergic neurons. That is, the light-evoked synaptic current had a short latency (4.4 ms, Fig. 8g), and these eIPSCs were sensitive to the Na$^+$ channel blocker tetrodotoxin (TTX, 1 μM), which abolished action potentials propagating to the synaptic terminals so that the synaptic transmission induced by light-evoked action potentials was blocked. However, in the presence of the K$^+$ channel blocker 4-aminopyridine (4-AP, 100 μM), which reduces membrane leakage current and thus enables the direct depolarization of synaptic terminals expressing ChR2 by light stimulation[18,19], TTX failed to block the light-evoked synaptic current in LPBN$^{CaMKII\alpha}$ neurons (Fig. 8e). These findings, together with the result that the TTX-insensitive

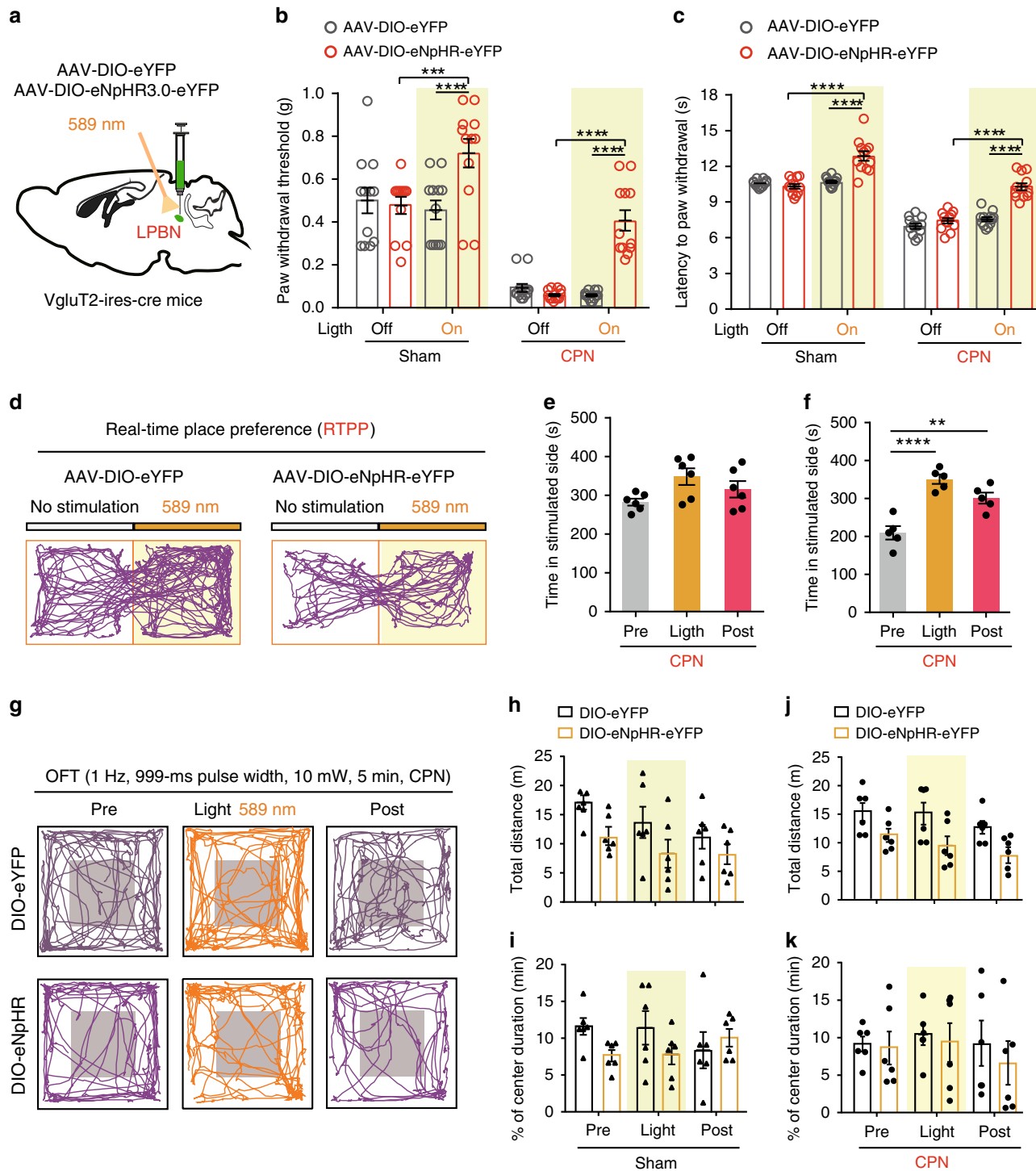

component was blocked by the GABA_A receptor antagonist bicuculline (10 µM) (Fig. 8e), further confirmed the monosynaptic innervation of LPBN$^{CaMKIIα}$ neurons by local GABAergic neurons.

To determine whether activation of GABAergic LPBN neurons functionally modulates glutamatergic LPBN neurons, LPBN$^{CaMKIIα}$ neurons expressing mCherry in brain slices were recorded with a lower intra-pipette Cl⁻ concentration (6 mM) under current-clamp with a membrane potential held around –45 mV so that spontaneous spikes could be detected (Supplementary Fig. 6c). We found that brief light stimulation (1 ms pulses at 20

Hz for 5 s) of GABAergic neurons expressing ChR2 significantly inhibited the spike frequency of LPBN$^{CaMKIIα}$ neurons for >10 s, accompanied by membrane hyperpolarization (Fig. 8h–k), indicating inhibited activity of LPBN$^{CaMKIIα}$ neurons mediated by local GABAergic neurons. We also found that CPN treatment increased the paired-pulse ratio of IPSCs induced by light stimulation of GABAergic LPBN neurons (Supplementary Fig. 6d, e), suggesting a decreased release probability of GABAergic synaptic vesicles[57]. On the other hand, the density of GABAergic LPBN neurons was not affected by CPN treatment (Supplementary Fig. 6a, b).

**Fig. 5 Photoinhibition of LPBN^VgluT2 neurons alleviates both basal nociception and neuropathic pain-like hypersensitivity. a** Schematic of illumination and virus injection of AAV-DIO-eYFP and AAV-DIO-eNpHR-eYFP in the LPBN of VgluT2-ires-Cre mice. **b** Yellow illumination (589 nm, 1 Hz, 999-ms pulse width, 10 mW) of LPBN^Vglut2 neurons significantly elevates the PWT in response to mechanical stimulation in both Sham-operated and CPN-ligated VgluT2-ires-Cre mice transfected with AAV-DIO-eNpHR virus, but not in mice transfected with DIO-eYFP virus. **c** Yellow illumination (589 nm, 1 Hz, 999-ms pulse width, 10 mW) of the LPBN significantly increases the latency of the thermal paw-withdrawal response in the Hargreaves test in both Sham-operated and CPN-ligated VgluT2-ires-Cre mice transfected with AAV-DIO-eNpHR virus, but not in mice transfected with DIO-eYFP virus. **d** Representative tracks of RTPP illustrating yellow (589 nm) light-evoked behavioral preference in CPN-ligated VgluT2-ires-Cre mice injected with AAV-DIO-eYFP (left) or DIO-eNpHR (right) virus in the LPBN. **e, f** Quantification of RTPP in CPN-ligated mice before (Pre), during (Light), and after (Post) 10-min yellow illumination of the LPBN in VgluT2-ires-Cre mice injected with AAV-DIO-eYFP (**e**) or AAV-DIO-eNpHR (**f**) virus. **g** Representative tracks in the open field test (OFT) before (Pre), during (Light), and after (Post) 5-min yellow laser illumination (589 nm, 1 Hz, 999-ms pulse width, 10 mW) of LPBN^Vglut2 neurons transfected with DIO-eYFP (upper) or DIO-eNpHR (lower) virus in CPN-ligated VgluT2-ires-Cre mice. **h–k** Quantification of total distance moved (**h, j**) and ratio of time spent in the periphery and center (**i, k**) in the OFT in VgluT2-ires-Cre mice transfected with DIO-eYFP or DIO-eNpHR virus in Sham-operated ((**h, i**) and CPN-ligated (**j, k**) mice. All data are presented as mean ± s.e.m. and error bars represent s.e.m. *P < 0.05, **P < 0.01, and ****P < 0.0001. See also Supplementary Table 1 for further statistical information. Source data are provided as a Source Data file.

**Prolonged activation of glutamatergic LPBN neurons induces chronic hyperalgesia.** It has been suggested that neuropathic pain results from plastic changes in the structure and function of the pain processing pathway induced by prolonged nociceptive signaling[1,6]. Since the photoactivation of glutamatergic LPBN neurons instantly induced hyperalgesia, we next investigated whether prolonged activation of these neurons can induce persistent pain, mimicking the development of neuropathic pain. To do this, we induced repetitive pharmacogenetic activation of glutamatergic LPBN neurons by daily intraperitoneal injection of CNO (1.5 mg kg$^{-1}$) for a week in VgluT2-ires-Cre mice with bilateral injection of AAV-DIO-hM3Dq-mCherry (hM3Dq) or AAV-DIO-mCherry (mCherry) into the LPBN (Fig. 9a). We found that daily CNO injection for one week induced a persistent decrease in the PWT for more than one month in mice expressing hM3Dq but not in those expressing mCherry (Fig. 9b), indicating that prolonged activation of glutamatergic LPBN neurons is sufficient to induce the development of chronic pain-like hypersensitivity.

We further determined whether this prolonged pharmacogenetic activation of glutamatergic LPBN neurons is sufficient to induce an aversive affective memory using the conventional conditioned place aversion (CPA) behavioral paradigm. Mice with hM3Dq or mCherry expression in glutamatergic LPBN neurons were habituated to two communicating chambers for 30 min on days 1 and 2. On day 3, the mice were tested for 10 min to confirm that there was no preference for one of the chambers. From day 4, the mice were injected (i.p.) with saline and paired to one side of the chamber for 30 min, and 4 h later, the mice were injected with CNO and paired to the other side for 30 min. The saline and CNO were injected once per day for 3 days. Twenty-four hours after the last injection of CNO, the mice were free to move to both sides, and the time in each chamber was measured (Fig. 9c). We found that the CNO injection induced significant place aversion for the CNO-paired chamber in mice transfected with hM3Dq, but not in those transfected with mCherry in the LPBN (Fig. 9d, e).

To determine whether this pharmacogenetic-induced chronic hyperalgesia can also be alleviated by the activation of GABAergic LPBN neurons, we simultaneously injected two types of virus into the LPBN of GAD2-ires Cre mice: AAV-CaMKIIα-hM3Dq-mCherry for the repetitive pharmacogenetic activation of glutamatergic LPBN neurons by daily CNO injections for a week, and Cre-dependent AAV-DIO-ChR2-eYFP for the instant optogenetic activation of GABAergic neurons (Fig. 9f). We found that the optogenetic activation of GABAergic LPBN neurons transfected with DIO-ChR2 transiently reversed the mechanical allodynia induced by CNO injection in mice transfected with CaMKIIα-hM3Dq in

glutamatergic LPBN neurons (Fig. 9g, red line, and day 21). Taken together, these results indicate that prolonged activation of glutamatergic LPBN neurons alone is sufficient to induce fully-developed chronic hyperalgesia that mimics CPN ligation-induced neuropathic pain-like hypersensitivity.

**Prolonged activation of GABAergic LPBN neurons prevents the development of neuropathic pain-like hypersensitivity.** The results that CPN ligation activated glutamatergic but not GABAergic LPBN neurons and that the prolonged selective pharmacogenetic activation of glutamatergic LPBN neurons was sufficient to induce chronic pain suggested that an imbalance between the excitatory and inhibitory activity of neurons in the LPBN may be responsible for the development of neuropathic pain-like hypersensitivity. Since local GABAergic neurons monosynaptically control glutamatergic LPNB neuronal activity and the activation of GABAergic LPBN neurons instantly alleviates neuropathic pain-like hypersensitivity, we next investigated whether prolonged activation of GABAergic LPBN neurons during the early stage (first week) of CPN ligation affects the development of neuropathic pain-like hypersensitivity. To do this, AAV-DIO-hM3Dq-mCherry or AAV-DIO-mCherry was bilaterally injected into the LPBN of GAD2-ires-Cre mice. Three weeks after virus infection, the mice received CPN ligation surgery, followed by daily CNO injections (i.p) for a week (Fig. 9h). We were surprised to find that CNO injection completely blocked the development of CPN ligation-induced neuropathic pain-like hypersensitivity in mice expressing hM3Dq, whereas CNO injection in control mice transfected with mCherry had no effect on the CPN-induced neuropathic pain-like hypersensitivity for up to 4 weeks, as assayed by the PWT (Fig. 9i). Consistently, we found that prolonged pharmacogenetic activation of GABAergic LPBN neurons reduced the CPN ligation-induced c-Fos expression (Fig. 9j, k). These results demonstrate that prolonged activation of GABAergic LPBN neurons during the first week of CPN ligation prevents the development of neuropathic pain-like hypersensitivity.

**Discussion**

Our study, using both gain-of-function and loss-of-function approaches, demonstrates that an LPBN circuit modulates both the sensory and emotional components of neuropathic pain-like hypersensitivity, consistent with the hypothesis that the LPBN not only functions as a relay nucleus for transmitting nociceptive signals from the spinal cord to higher centers, but also actively participates in the development of neuropathic pain, a persistent unpleasant experience involving both sensory and aversive emotional components. In this respect, it is interesting to note

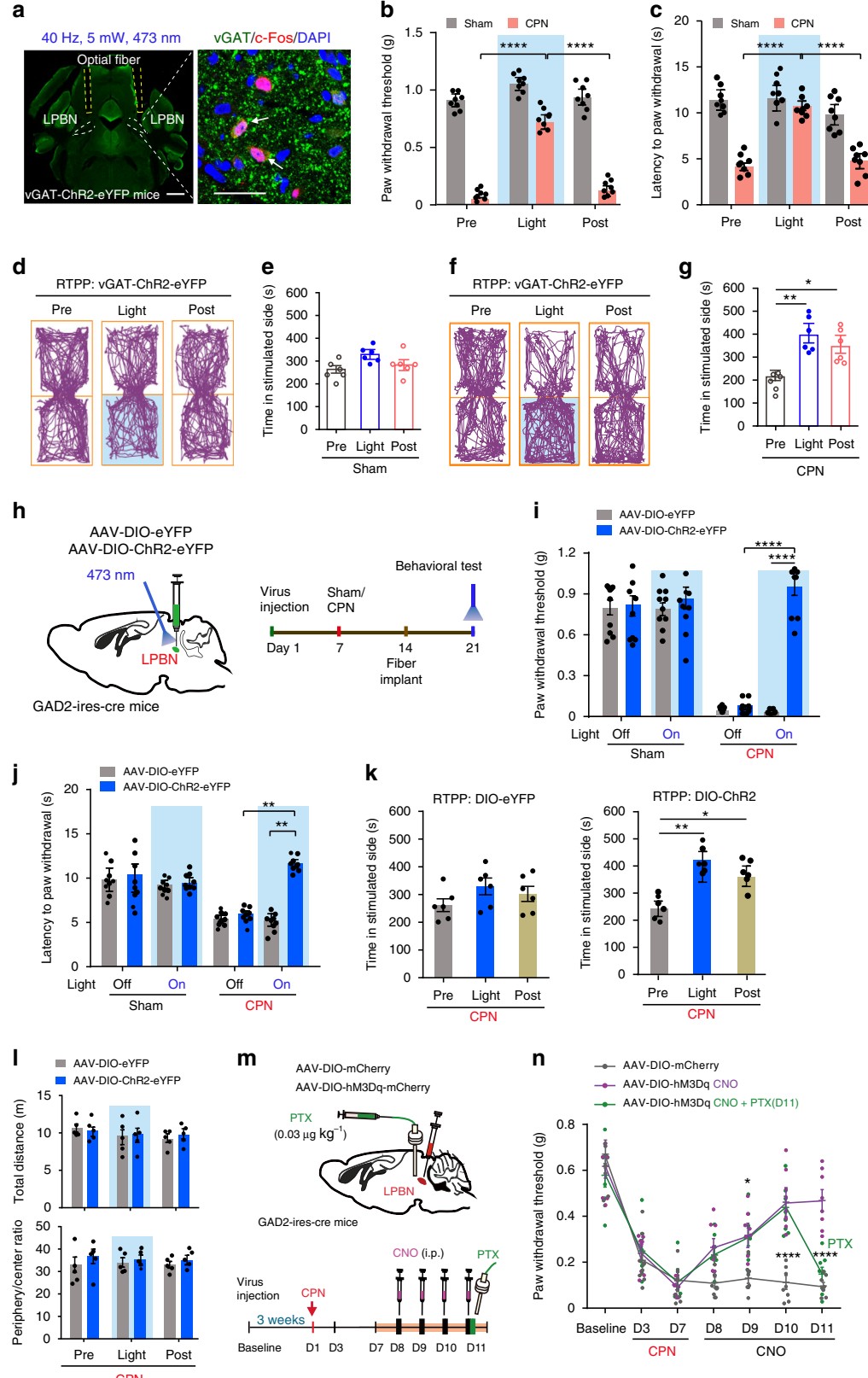

that the PBN has also been shown to regulate various aversive emotional behaviors including threat and avoidance memory[31,58], anorexia[30,33,34,54], and escape behavior in response to noxious stimuli[59,29].

**Activation of glutamatergic LPBN neurons is necessary and sufficient to induce neuropathic pain-like behavior.** We found here that glutamatergic LPBN neurons were selectively activated in neuropathic pain-like hypersensitivity, as evidenced by

**Fig. 6 Optogenetic stimulation of LPBN GABAergic neurons suppresses neuropathic pain-like hypersensitivity, but not basal nociception. a** Image showing optic fibers position in the LPBN (left; scale bar, 1 mm) and c-Fos (red) co-labeled with vGAT-positive neurons (green) in the LPBN after optogenetic stimulation (right; arrows, double-labeled neurons; scale bar, 30 μm). **b, c** Photoactivation of GABAergic LPBN neurons elevate the PWT (**b**) and thermal paw-withdrawal latency (**c**) in CPN-ligated, but not in Sham-operated vGAT-ChR2-eYFP mice. **d, f** Tracking maps of RTPP from a Sham (**d**) and a CPN-ligated mouse (**f**) before, during, and after optogenetic stimulation of GABAergic LPBN neurons. **e, g** Quantification of time spent in stimulated side as shown in **d** and **f**. **h** Schematic of virus delivery (left) and timeline of the behavioral experiments (right). **i, j** Optogenetic activation of the GABAergic LPBN neurons increased the PWT (**i**) and thermal paw-withdrawal latency (**j**) in CPN-ligated mice, but not in Sham-operated mice or in mice transfected with DIO-eYFP. **k** Quantification of time spent in the non-preferred RTPP chamber before (Pre), during (Light), and after (Post) laser stimulation of CPN-ligated GAD2-ires-Cre mice transfected with AAV-DIO-eYFP (left) or AAV-DIO-ChR2-eYFP (right). **l** Quantification of total distance moved (upper) and ratio of time spent in the periphery and center (lower) in the OFT before (Pre), during (Light), and after (Post) light stimulation of GABAergic LPBN neurons in GAD2-ires-Cre mice. **m** Experimental design for pharmacogenetic activation of GABAergic LPBN neurons. **n** Time-course of the CPN ligation-induced decrease in PWT and the effect of pharmacogenetic activation of GABAergic LPBN neurons which was reversed by administration of PTX (0.03 μg kg$^{-1}$) via a cannula into the LPBN at day 11. All data are presented as mean ± s.e.m. and error bars represent s.e.m. *$P < 0.05$, **$P < 0.01$, and ****$P < 0.0001$. See also Supplementary Table 1 for further statistical information. Source data are provided as a Source Data file.

increased c-Fos expression (Fig. 1a–d) and the noxious pinch and mechanical stimulation-induced dramatic Ca$^{2+}$ elevation in glutamatergic (Fig. 1j, k and Supplementary Fig. 2d–g) but not in GABAergic neurons (Fig. 1m, n) in CPN-ligated mice. Interestingly, the optogenetic activation of glutamatergic LPBN neurons instantly induced neuropathic pain-like symptoms in naïve mice, including mechanical allodynia (Fig. 2d and Fig. 3b) and thermal hyperalgesia (Fig. 2e and Fig. 3c), as well as RTPA behavior (Fig. 2g–j and Fig. 3d, e), whereas the optogenetic inhibition of glutamatergic LPBN neurons instantly relieved neuropathic pain-like hypersensitivity in CPN-ligated mice (Fig. 4b, c and Fig. 5b, c). These results indicate that the activation of glutamatergic LPBN neurons is both necessary and sufficient for the induction and maintenance of neuropathic pain-like behavior. We found that optogenetic activation of LPBN$^{VgluT2}$ neurons did not further enhance hypersensitivity in CPN-ligated mice (Fig. 2d, e). It is possibly that CPN treatment already induced hypersensitivity to the maximum extent, which would mask the effects of optogenetic activation of LPBN$^{VgluT2}$ neurons. These results suggest that the two types of treatments (optogenetic activation of LPBN$^{VgluT2}$ neurons and CPN treatment) share the same mechanisms for inducing hypersensitivity.

We found that prolonged pharmacogenetic activation of glutamatergic LPBN neurons for one week resulted in fully-developed neuropathic pain-like hypersensitivity that persisted for several weeks after the termination of pharmacogenetic activation (Fig. 9a, b). This is particularly interesting, since neuropathic pain is characterized as spontaneous discomfort or as painful hypersensitivity to temperature and touch that persists even after recovery from nerve injury[1,2,7]. Structural and functional plasticity induced by nerve trauma leading to the sensitization and increased excitability of peripheral nociceptors or central neurons, especially in the spinal dorsal horn[1–3,5,17,60,61], have been implicated in the post-injury hypersensitivity. Our findings suggest that the plasticity and sensitization of glutamatergic LPBN neurons induced by prolonged nociceptive signaling following nerve injury is also crucial for the development of persistent neuropathic pain.

The results that optogenetic inhibition of glutamatergic LPBN neurons not only blocked neuropathic pain-like hypersensitivity but also elevated the basal nociception threshold in naïve mice (Fig. 4b, c, and Fig. 5b, c) are consistent with the hypothesis that glutamatergic LPBN neurons are crucial for both relaying physiological pain and mediating neuropathic pain.

**GABAergic LPBN neurons selectively gate neuropathic pain-like behavior but not basal nociception.** It is interesting to note that GABAergic neurons only constitute ~10.1% of LPBN neurons, much less than that reported in the spinal dorsal horn (~70%)[62] and in the central nucleus of the amygdala (~50.78%), but comparable to that reported in the thalamus (~10.43%) and somatosensory cortex (~8.66%) (Allen Brain Atlas, http://www.brain-map.org) in the pain transmission pathway, suggesting that glutamatergic LPBN neurons that also receive GABAergic innervation from other regions[28,54,53] are under the relatively weak control of local GABAergic neurons. Consistent with this proposal, we found that the optogenetic activation of GABAergic LPBN neurons did not affect the paw-withdraw response in naïve mice (Fig. 6b, c). However, we found that glutamatergic LPBN neurons receive direct monosynaptic input from local GABAergic neurons, as demonstrated by monosynaptic virus tracing and direct electrophysiological recording (Fig. 8a–k). Interestingly, optogenetic activation of GABAergic LPBN neurons efficiently blocked the mechanical allodynia, thermal hyperalgesia, and affective pain-like behavior in CPN-ligated mice (Fig. 6b–l). Furthermore, the analgesia induced by pharmacogenetic activation of GABAergic LPBN neurons in CPN-ligated mice was reversed by injection of the GABA$_A$ receptor chloride channel blocker PTX into the LPBN, suggesting that GABAergic LPBN neurons gate neuropathic pain transmission via local inhibition of glutamatergic neurons (Fig. 6m, n). However, our study does not exclude the possibility that GABAergic input from other brain regions to the LPBN may also play an important role in neuropathic pain-like hypersensitivity. In this respect, it should be noted that a recent study has shown that suppressed GABAergic input from the central nucleus of the amygdala to the LPBN may contribute to chronic pain[28].

The result that optogenetic activation of GABAergic LPBN neurons did not affect the paw-withdrawal response in naïve mice is in sharp contrast to the effect of direct optogenetic inhibition of glutamatergic LPBN neurons, which not only blocked neuropathic pain-like hypersensitivity but also elevated the basal nociception threshold in naïve mice (Fig. 4b, c and 5b, c). One possibility for the discrepancy between the result of optogenetic inhibition of glutamatergic LPBN neurons and that of optogenetic activation of GABAergic LPBN neurons may be that the former, relaying physiological pain, may receive relatively weak inhibition from local GABAergic neurons, while the development of neuropathic pain may recruit an additional population of glutamatergic neurons that are normally under tighter control by local GABAergic neurons. In this way, the basic physiological pain signal important for survival can be faithfully relayed without spreading to and sensitizing the glutamatergic LPBN neurons involved in pathological pain. Consistent with this hypothesis, we found that optogenetic inactivation of GABAergic LPBN neurons was sufficient to induce hyperalgesia and RTPA in control mice

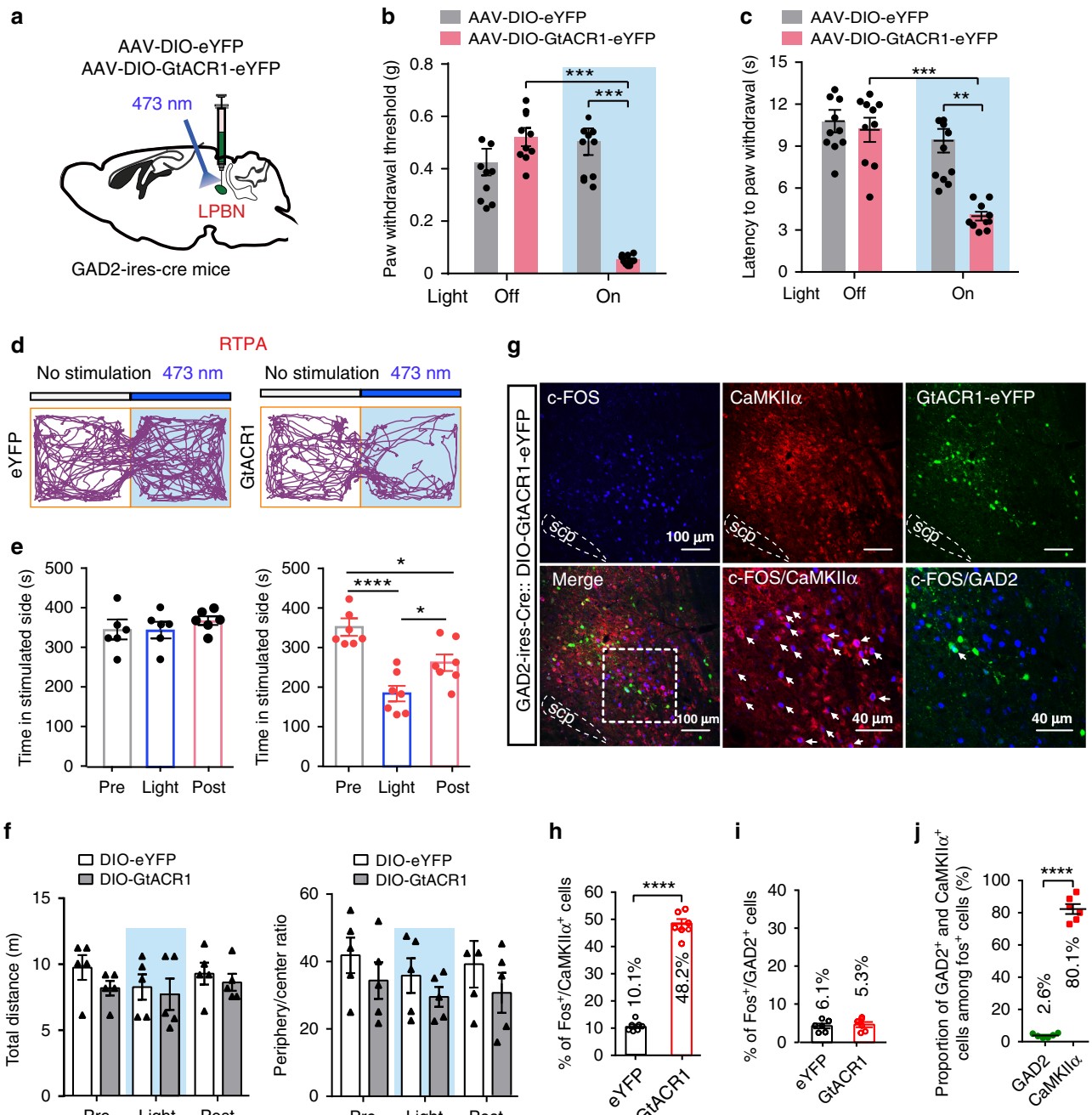

**Fig. 7 Optogenetic inhibition of GABAergic LPBN neurons induced neuropathic pain-like symptoms. a** Schematic of the stereotaxic delivery of AAV-DIO-GtACR1-eYFP and AAV-DIO-eYFP into the LPBN of GAD2-ires-Cre mice and photoinhibition of GABAergic LPBN neurons. **b** Photoinhibition (473 nm, 40 Hz, 5 mW) of GABAergic LPBN neurons decreases the PWT in GAD2-ires-Cre mice transfected with AAV-DIO-GtACR1-eYFP. **c** Photoinhibition of GABAergic LPBN neurons decreases thermal withdraw latency in GAD2-ires-Cre mice transfected with AAV-DIO-GtACR1-eYFP. **d** Representative tracks in RTPA during 10-min illumination of the LPBN in GAD2-ires-Cre mice transfected with AAV-DIO-eYFP (left) or AAV-DIO-GtACR1-eYFP (right). **e** Quantification of time spent in the preferred chamber in the RTPA test before (Pre), during (Light), and after (Post) 10-min illumination of the LPBN in GAD2-ires-Cre mice transfected with AAV-DIO-eYFP (left) or AAV-DIO-GtACR1-eYFP (right). **f** Quantification of total distance moved (left) and ratio of time spent in the periphery and center (right) in the OFT before (Pre), during (Light), and after (Post) 10-min illumination of GABAergic LPBN neurons in GAD2-ires-Cre mice transfected with AAV-DIO-eYFP or AAV-DIO-GtACR1-eYFP. **g** Representative images of c-Fos expression (blue) 1 h after optogenetic inhibition (473 nm, 5 mW, 10 min) of GABAergic LPBN neurons in GAD2-ires-Cre mice transfected with AAV-DIO-GtACR1-eYFP. Lower middle and lower right panels show magnified views of the boxed area in the lower left panel. Note that c-Fos-positive cells co-localize with CaMKIIα (red), but not with GtACR1-eYFP (GAD2, green). Arrows indicate CaMKIIα[+] neurons co-labeled with c-Fos. Scale bars, 100 μm and 40 μm (lower middle and lower right panels). **h, i** Quantification of cells expressing c-Fos induced by illumination of GABAergic LPBN neurons in GAD2-ires-Cre mice transfected with AAV-DIO-GtACR1-eYFP or AAV-DIO-eYFP as in **g**. **j** Proportion of c-Fos-positive cells that co-express GAD2-eYFP or CaMKIIα[+] cells 1 h after optogenetic inhibition of GABAergic LPBN neurons as shown in **g**. All data are presented as mean ± s.e.m. and error bars represent s.e.m. *$P < 0.05$, **$P < 0.01$, ***$P < 0.001$, and ****$P < 0.0001$. See also Supplementary Table 1 for further statistical information. Source data are provided as a Source Data file.

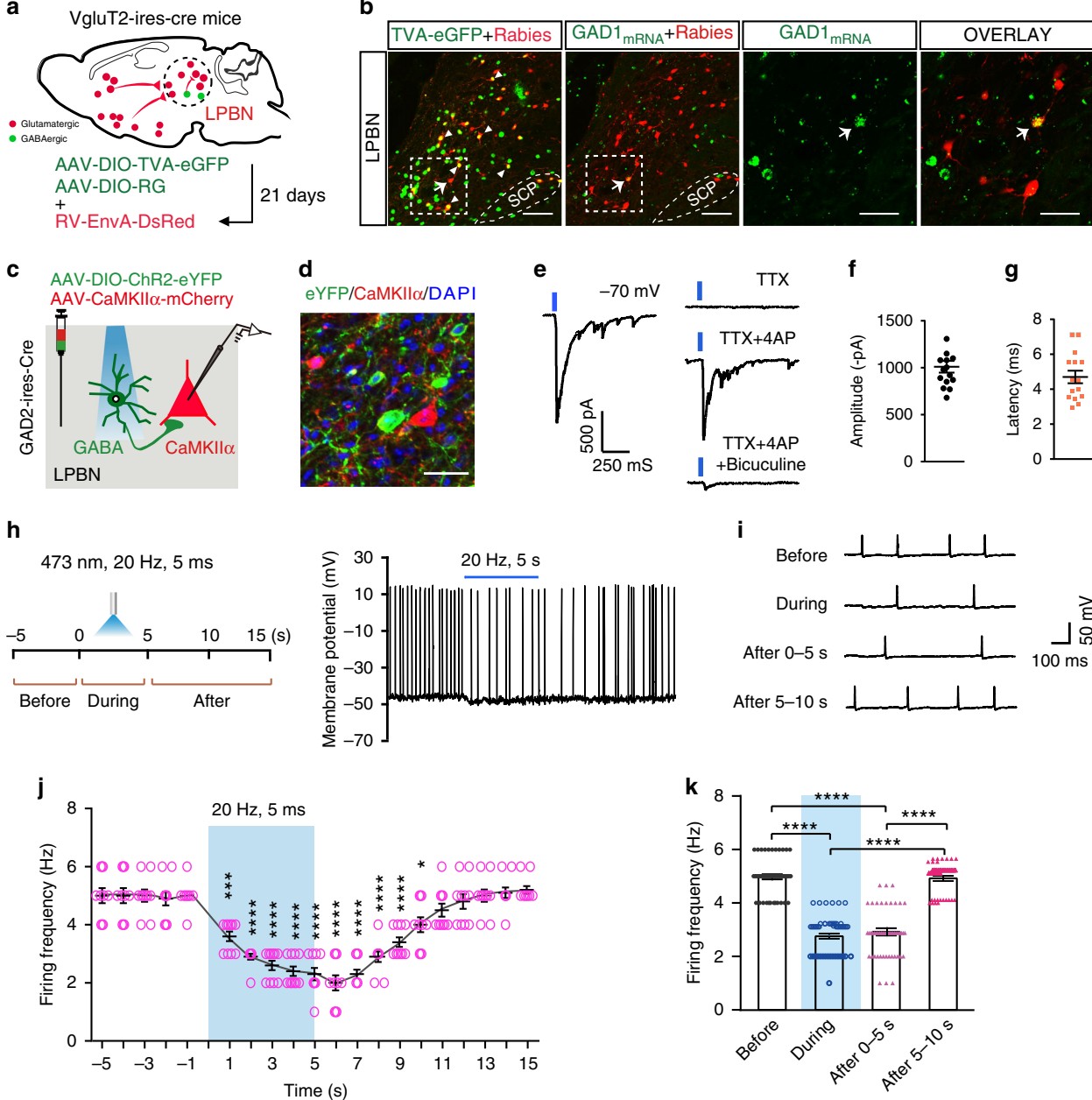

**Fig. 8 GABAergic LPBN neurons monosynaptically innervate local glutamatergic neurons. a** Schematic of monosynaptic retrograde rabies virus tracing from glutamatergic LPBN neurons. **b** Representative images showing monosynaptic rabies spread from glutamatergic LPBN neurons in VgluT2-ires-Cre mice. Arrowheads, starter neurons infected by both TVA-eGFP (green) and rabies virus (red). The arrow indicates a GABAergic neuron (labeled with GAD1 mRNA, green) that was trans-synaptically infected by rabies virus. The third and fourth panels from the left show magnified views of the boxed areas in the second panel. Scale bars, 100 μm and 30 μm (magnified views). **c** Schematic of patch clamp recording from glutamatergic LPBN neurons transfected with AAV-CaMKIIα-mCherry in response to light stimulation of GABAergic LPBN neurons transfected with AAV-DIO-ChR2-eYFP. **d** Example image showing the expression of ChR2 in GABAergic neurons (green) and mCherry in CaMKIIα neurons (red) for targeted whole-cell patch-clamp recordings as in **c**. Scale bar, 30 μm. **e** Example of light-evoked IPSCs recorded from mCherry-positive glutamatergic neurons at a holding potential of –70 mV (left panel). The IPSCs were blocked by 1 μM TTX, and they were rescued by 100 μM 4-AP and blocked again by 10 μM bicuculline (right panel). $n = 12$ cells in 6 slices from 4 mice. **f, g** Averaged amplitude and latency of recorded IPSCs as in **o**. $n = 15$ traces recorded in 6 slices from 4 mice. **h** Workflow (left) and representative current clamp recording trace (right) of spontaneous spikes in an mCherry-positive LPBN$^{CaMKIIα}$ neuron at a holding potential of –45 mV. **i** Extended traces from (**h**) with a slow time scale showing light-induced changes in spike frequency of an LPBN$^{CaMKIIα}$ neuron sampled before, during, and after light stimulation of GABAergic LPBN neurons. **j** Time course of summarized changes in spike frequency of LPBN CaMKIIα neurons induced by light stimulation of GABAergic LPBN neurons. **k** Averaged firing frequency of LPBN CaMKIIα neurons during each period as indicated. All data are presented as mean ± s.e.m. and error bars represent s.e.m. $*P < 0.05$, $**P < 0.01$, and $****P < 0.0001$. See also Supplementary Table 1 for further statistical information. Source data are provided as a Source Data file.

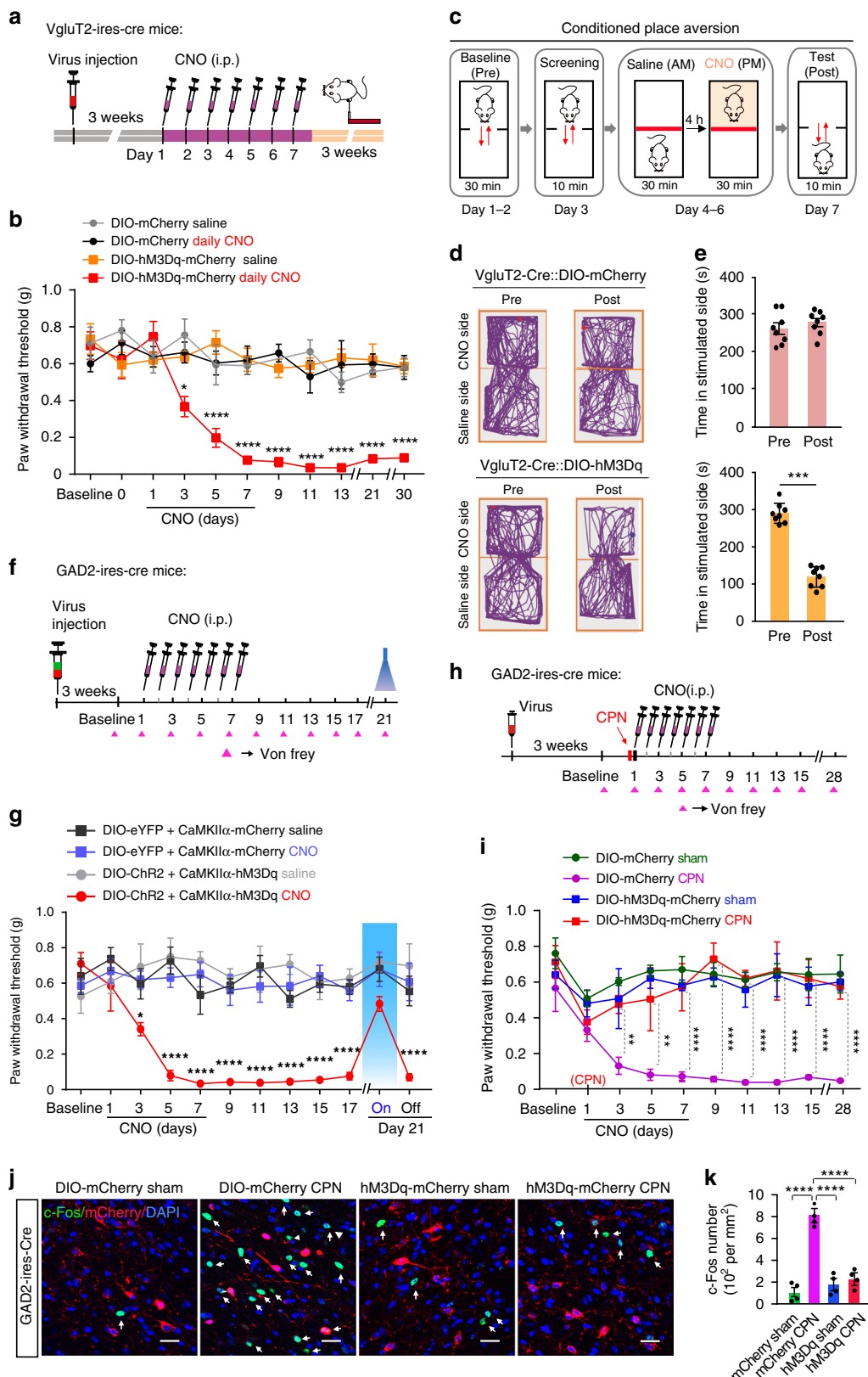

mimicking neuropathic pain (Fig. 7a–e). It is interesting to note that the activation of hunger-sensitive and PBN-projecting arcuate nucleus neurons that co-express agouti-related protein, neuropeptide Y, and GABA attenuates inflammatory pain-like hypersensitivity without altering acute nociceptive responses[33].

**Development of neuropathic pain-like hypersensitivity is due to an imbalance of excitatory-inhibitory neuronal activity in the LPBN circuit.** We found that most of the cells expressing c-Fos induced by CPN ligation (~74.8%) were glutamatergic (constituting ~41.8% of all glutamatergic neurons), while only

**Fig. 9 Persistent activation of glutamatergic and GABAergic LPBN neurons, respectively, develops and prevents neuropathic pain-like hypersensitivity. a** Experimental design and timeline of the behavioral experiment. **b** Time-course of PWT changes induced by daily injection of CNO (i.p., 1.5 mg kg$^{-1}$) for one week in VgluT2-ires-Cre mice. **c** Schematic of the experiment design for conditioned place aversion (CPA). **d** Examples of tracking maps in the CPA before (Pre) and after (Post) injection of CNO (i.p., 1.5 mg kg$^{-1}$) into the LPBN of VgluT2-ires-Cre mice transfected with DIO-mCherry (upper) or DIO-hM3Dq-mCherry (lower). **e** Quantification of time spent in the preferred chamber as in **d. f** Experimental design and timeline of the behavioral experiment in GAD2-ires-Cre mice simultaneously transfected with two types of virus in the LPBN, one carrying DIO-eYFP (control) or DIO-ChR2-eYFP and the other carrying CaMKIIα-mCherry (control) or CaMKIIα-hM3Dq-mCherry. **g** Time-course of PWT changes induced by daily injection of CNO (i.p., 1.5 mg kg$^{-1}$) for one week in GAD2-ires-Cre mice. A persistent decrease in PWT was induced only in mice injected with CNO and transfected with CaMKIIα-hM3Dq-mCherry, but not in the other groups. Note that illumination (473 nm, 40 Hz, 5 mW) of GABAergic LPBN neurons on day 21 reversibly elevated the PWT in mice transfected with DIO-ChR2-eYFP. **h** Experimental design and timeline of the behavioral experiment. **i** Daily injection of CNO for one week initiated on the same day as CPN ligation prevented the development of neuropathic pain-like hypersensitivity in mice transfected with DIO-hM3Dq-mCherry (red), but not in mice transfected with DIO-mCherry. **j** Representative images showing c-Fos expression (green, arrows) and GABAergic neurons transfected with mCherry or hM3Dq-mCherry (red) in the LPBN from Sham-operated and CPN-ligated mice. Scale bar, 30 μm. Blue, DAPI stain). **k** Quantification of c-Fos-positive cells in the LPBN from different groups as in **j**. All data are presented as mean ± s.e.m. and error bars represent s.e.m. *$P < 0.05$, **$P < 0.01$, and ****$P < 0.0001$. See also Supplementary Table 1 for further statistical information. Source data are provided as a Source Data file.

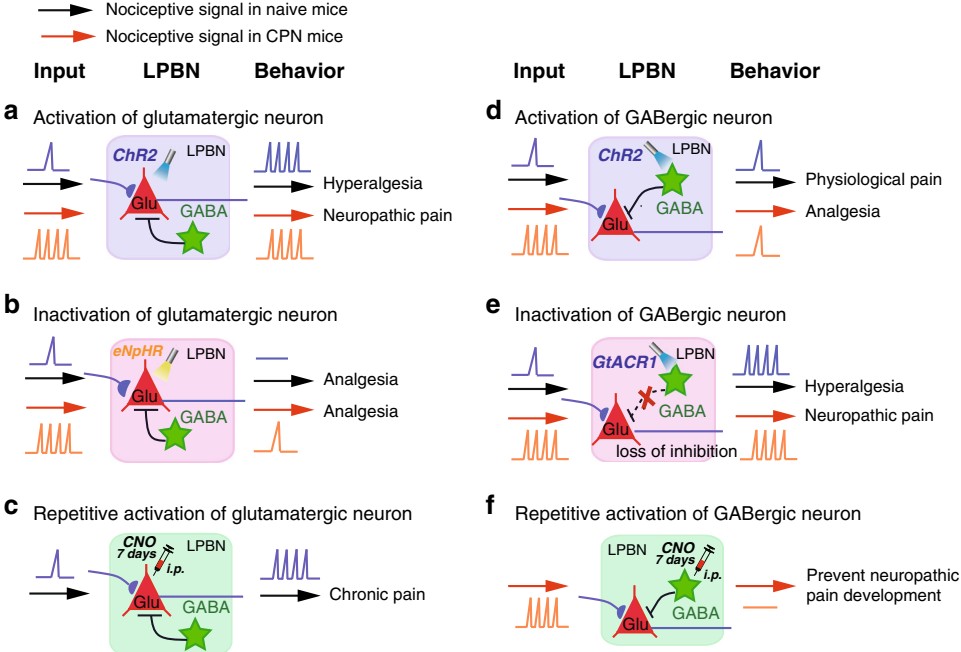

**Fig. 10 Schematic summary of the main findings illustrating how activity in the LPBN circuit governs physiological and pathological pain-like hypersensitivity. a** Light-activation of glutamatergic neurons induces hyperalgesia in naive mice to an extent similar to that in CPN-ligated mice. **b** Light-inactivation of glutamatergic neurons induces analgesia in both naive and CPN-ligated mice. **c** Daily pharmacogenetic activation of glutamatergic neurons for one week induces chronic nociception, mimicking neuropathic pain. **d** Light-activation of GABAergic neurons inhibits neuropathic pain-like hypersensitivity, but not basal nociception. **e** Light-inactivation of GABAergic neurons induces hyperalgesia in naive mice to an extent similar to that in CPN-ligated mice. **f** Daily pharmacogenetic activation of GABAergic neurons for one week prevents the development of neuropathic pain-like hypersensitivity in CPN-ligated mice.

2.1% of the c-Fos-positive cells were GABAergic (constituting ~8.5% of all GABAergic neurons). These results, together with the finding that after CPN ligation, pinch stimulation induced dramatic Ca$^{2+}$ elevation in glutamatergic, but not GABAergic neurons (Fig. 1i–n), suggest that glutamatergic neurons are selectively activated and sensitized by CPN ligation. Thus, the robust nociceptive signaling from the spinal cord due to prolonged noxious stimulation such as CPN ligation may upset the excitatory-inhibitory balance for a sufficient time within the LPBN circuit and lead to the development of neuropathic pain-like hypersensitivity (Fig. 10).

The results that optogenetic activation of GABAergic LPBN neurons instantly relieved neuropathic pain-like hypersensitivity (Fig. 6b, c, i, j) and that prolonged pharmacogenetic activation of

these neurons at an early stage of CPN ligation completely blocked the development of neuropathic pain-like hypersensitivity (Fig. 9i) indicate that they not only control the transmission but also govern the development of neuropathic pain. Prolonged activation of GABAergic LPBN neurons may restore the balance of excitation and inhibition in the LPBN circuit and prevent the sensitization of glutamatergic neurons induced by CPN ligation (Fig. 10). Indeed, we found that repetitive pharmacogenetic activation of GABAergic LPBN neurons prevented the increased c-Fos expression in the LPBN induced by CPN ligation (Fig. 9j, k). It has been reported that the development (early phase) and maintenance (late phase) of neuropathic pain may have distinct mechanisms[63–65]. It will be interesting to determine whether GABAergic LPBN neurons also play a critical role in the

late phase of neuropathic pain-like hypersensitivity (after 3 weeks).

The densely-distributed GABAergic and glycinergic interneurons in the dorsal horn have been regarded as a key component of the gate control theory of pain[12,60,66,67]. Disinhibition due to the selective loss of GABAergic interneurons or collapse of the Cl⁻ gradient in the dorsal horn neurons mediated by glial cell-derived BDNF and cytokines has been proposed as a mechanism of opening the gate for pain[12,68–70]. It is unclear if glial cells and cytokines also play a role in the LPBN in neuropathic pain. We found that CPN treatment did not change the density of GABAergic LPBN neurons, although the release probability of these neurons decreased (Supplementary Fig. 6a–e). Our study demonstrates that the balance of excitation and inhibition in the LPBN circuit may provide another important gate control of neuropathic pain. Unlike in the dorsal horn, however, GABAergic neurons constitute only a very small population in the LPBN. Due do the relatively weak GABAergic control of the LPBN circuit, selective activation of glutamatergic over GABAergic neurons in the LPBN together with a decreased release probability of GABAergic LPBN neurons by nerve injury is thus sufficient to break this delicate excitatory-inhibitory balance, leading to loss of gate control and the development of neuropathic pain (Fig. 10).

Sex dimorphism has been reported in some pain mechanisms[71–73]. In particular, a recent study reported that neuropathic pain is accompanied by sex-specific changes in the excitability and synaptic function of different subtypes of GABAergic neurons in prelimbic cortex[72]. We only used male mice here and it would be interesting to determine in future whether similar sex-specific changes occur in LPBN neurons in neuropathic pain-like hypersensitivity.

The intricate relationship between pain and affective and emotional responses has been well studied, and the amygdala, which receives nociceptive signals directly from the PBN[18,74,75], is regarded as a key node in processing the affective and emotional components of pain[4,14,35,76]. The sustained nociceptive signaling caused by nerve injury may result in structural and functional plasticity in the spinoparabrachial pathway. The sensitized glutamatergic LPBN neurons may then convey sensory signals to the amygdala with heightened sensitivity and associate the neuropathic pain with aversive emotional responses (Fig. 10). It is also possible that LPBN modulates neuropathic pain through activation of descending modulation pathways, such as basolateral amygdala-prefrontal cortex-periaqueductal gray (PAG)-spinal cord pathway reported recently[77].

Neuropathic pain remains a major public health burden and effective therapy is still lacking[78]. Since neuropathic pain also happens in in supraspinal CNS injury or brain disorders[1,2], higher brain regions also play crucial roles in neuropathic pain development. Our findings suggest that the LPBN is a potential key target for neuropathic pain therapy.

## Methods

**Animals**. In all experiments, vGAT-ChR2 (H134R)-eYFP (vGAT) (JAX014548), VgluT2-ires-Cre (Slc17a6tm2(cre)Lowl/J) (JAX016963), GAD2-ires-Cre (B6N.Cg-Gad2tm2(cre)Zjh/J) (JAX019022), Gad67-GFP (from the Takeshi Kaneko laboratory of Kyoto University), and wild-type (C57BL/6J, from the Shanghai SLAC Laboratory Animal CO. LTD) male mice 8–10 weeks of age (20–35 g) were used. All experiments were conducted in accordance with the guidelines for the care and use of laboratory animals of Zhejiang University (ZJU) and were approved by the Animal Advisory Committee at ZJU. The mice were housed 4 or 5 per cage at constant room temperature (25 ± 1 °C) and relative humidity (60 ± 5%) under a 12 h light/dark schedule (light 07:00–19:00) with food and water ad libitum. All behavioral tests were carried out during the light phase. In the behavioral tests, the mice were allowed to adapt to laboratory conditions for ~1 week and to habituate to the testing situation for at least 30 min before experiments. Littermate mice were split into random groups before virus injections. Biohazard wastes were managed following the Medical Waste Management Regulations of China.

**CPN ligation surgeries**. The common peroneal nerve (CPN) ligation model of neuropathic pain has been used[39]. Briefly, mice were anesthetized with sodium pentobarbital (1% wt/vol). The left common peroneal nerve between the anterior and posterior groups of muscles was ligated slowly with 5-0 chromic gut suture (Ethicon) until twitching of the digits was observed. The skin was sutured with 5-0 silk suture and cleaned with Povidone-iodine. Sham surgery was conducted in the same manner, but the nerve was not ligated. All mice were kept in a normal living cage after surgery.

**Measurement of Hypersensitivity Threshold in Mice**. Mice should have 3 days of acclimation in all the behavior tests. Measurements of mechanical and thermal withdrawal threshold/latency were performed[43]. The Von Frey test was used to assess the onset and maintenance of mechanical allodynia over time[46]. Dynamic tactile allodynia testing was assessed by lightly stroking the lateral external side of the surface of the injured hindpaw, in the direction from heel to toe with a von Frey monofilament without bending the filament. After acclimatizing the mice for about 60 min in the observation chambers on top of a grid mesh, the ipsilateral paws were tested 10 times, with intervals of 10–15 s, and the percentage of withdrawal for ipsilateral paw (nerve-injured side) was considered as the hypersensitivity response. Mice were tested before blue or yellow light stimulation (light Off) and after 45 min they were stimulated with blue or yellow light on (light On) for 3 min and the ipsilateral hindpaws were tested. The stimulus producing a 50% likelihood of a withdrawal response was determined and taken as the paw withdrawal threshold (PWT)[46,79].

The Hargreaves test is specifically designed to assess thermal pain sensation in rodents[80]. Each mouse was placed in a plastic cylinder on a glass floor. A radiant heat beam was focused onto the injured hindpaw (plantar test apparatus from IITC Life Science Inc.). The latency to hindpaw withdrawal of the ipsilateral paw (nerve-injured side) was recorded with at least 3 trials per animal repeated 5 min apart. Beam intensity was adjusted so that control mice displayed a latency of 8–12 s. A cut-off latency of 30 s was set to avoid tissue damage. Measurement of hypersensitivity threshold was performed during the light cycle by the same researcher who was blinded to the surgery, genotypes, and treatments (CNO versus saline).

**Stereotaxic injection**. To manipulate the activity of LPBN neurons, adeno-associated viruses (AAVs) were injected bilaterally into the LPBN (AP, 5.02 mm; ML, ± 1.25 mm; DV, 3.27 mm) of 8–10 week-old mice and behavioral tests were performed 2–3 weeks later. All injections were performed with a stereotaxic apparatus on mice under anesthesia with sodium pentobarbital (1% wt/vol), and body temperature maintained with a heating pad. Ophthalmic ointment was applied to maintain eye lubrication. Viruses were injected at 15 nl/min using an air pressure system connected to a glass pipette (tip diameter 10–30 μm). After injection, the pipette was left in place for 15–20 min to allow diffusion. The mice were allowed to recover from anesthesia on a heating blanket before returning to the home cage. Antibiotic (Ceftriaxone sodium, 0.1 g/kg) was injected intraperitoneally daily for three days after surgery to prevent infection.

**Viral injection and optical fibers/cannula implantation**. Mice were anesthetized with pentobarbital sodium (0.1 g/kg) and mounted in a stereotaxic apparatus. The skull was exposed by midline scalp incision, and a craniotomy was performed unilaterally or bilaterally for introduction of a microinjection glass pipette into the LPBN (AP, 5.02 mm; ML, ± 1.25 mm; DV, 3.27 mm) in a volume of 80–100 nl for each side at 0.05 μl min⁻¹. The syringe was not removed until 15–20 min after the end of infusion to allow diffusion of the virus. A craniotomy window (~1.5 mm diameter) over the target area was made using a hand-held drill[41,81].

For fiber photometry, 100 nl of AAV2/9-hSyn-DIO-GCaMP7s-WPRE (1.34 × 10¹³ genomic copies per ml) was unilaterally injected into the LPBN. After injection of AAV2/9-DIO-GCaMP7s, an optical fiber (outer diameter (OD) of 200 μm, numerical aperture (NA) of 0.37, purchased from Inper) was placed the injection site. For control fiber photometry experiments, 100 nl of AAV2/9-EF1α-DIO-eYFP (1.0 × 10¹² genomic copies per ml) was used instead of GCaMP7s.

For local optogenetic stimulation, 80–100 nl of AAV2/9-CaMKIIα-hChR2 (H134R)-mCherry (1.97 × 10¹³ genomic copies per ml), AAV2/8-EF1α-DIO-ChR2 (H134R)-mCherry (1.25 × 10¹² genomic copies per ml), AAV2/9-EF1α-DIO-eYFP (1.0 × 10¹² genomic copies per ml), AAV2/8-EF1α-DIO-mCherry (1.1 × 10¹² genomic copies per ml), AAV2/8-CaMKIIα-mCherry (2.13 × 10¹² genomic copies per ml) was unilaterally injected into the LPBN.

For local photoinhibition stimulation, AAV2/9-CaMKIIα-eNpHR3.0-mCherry (8.3 × 10¹² genomic copies per ml) or AAV2/8-CaMKIIα-mCherry (2.13 × 10¹² genomic copies per ml); AAV2/9-EF1α-DIO-eNpHR3.0-eYFP (1.3 × 10¹³ genomic copies per ml) or AAV2/9-EF1α-DIO-eYFP (1.0 × 10¹² genomic copies per ml); AAV2/9-EF1α-DIO-hGtACR1-P2A-eYFP-WPRE (5 × 10¹³ genomic copies per ml) or AAV2/9-EF1α-DIO-eYFP (1.0 × 10¹² genomic copies per ml) was bilaterally injected into the LPBN.

For pharmacogenetic stimulation, AAV2/9-hSyn-DIO-hM3Dq-mCherry (2.04 × 10¹² genomic copies per ml) or AAV2/8-EF1α-DIO-mCherry (1.1 × 10¹² genomic copies per ml); AAV2/9-CaMKIIα-hM3Dq-mCherry (4 × 10¹³ genomic copies per ml) or AAV2/8-CaMKIIα-mCherry (2.13 × 10¹² genomic copies per ml)

was bilaterally injected into the LPBN. These viruses were made by SunBio Biomedical technology Co. or Taitool Bioscience Co. Ltd (Shanghai, China).

For monosynaptic retrograde tracing, 60–80 nl of a mixture (1:1) of AAV2/9-CAG-DIO-TVA-eGFP ($1.7 \times 10^{13}$ genomic copies per ml) and AAV2/9-CAG-DIO-RG ($6.8 \times 10^{12}$ genomic copies per ml) was unilaterally injected into the LPBN. We injected the 80–100 nl EnvA-pseudotyped, glycoprotein (RG)-deleted and DsRed-expressing rabies virus (RV-EnvA-DsRed) ($5.0 \times 10^8$ genomic copies/ml) into the same site 3 weeks later (BrainVTA, China). For the behavioral experiments, animals were allowed to recover for 3–6 weeks, then they were handled and habituated to the behavioral environment every day for one week. For all experiments, mice with incorrect injection sites were excluded from further analysis.

For optogenetic manipulation in awake behaving mice, optical fibers (NA 0.37) were implanted to target the LPBN bilaterally (AP, 5.02 mm; ML, ± 1.57 mm; DV, 2.97 mm at an angle of 5°) two weeks after injection of AAV into the LPBN. The fibers (diameter, 200 μm; length: 6.0 mm) were attached to the skull with dental cement and connected to a laser source using an optical fiber sleeve. The power of the blue (473 nm, 5–10 mW) or yellow laser (589 nm, 1 Hz, 999 ms, 10 mW) (Newdoon Inc., Hangzhou, China) was 0.83–3.33 mW mm$^{-2}$ as measured at the tip of the fiber. GABA neurons were activated by blue light pulses (10 ms) at 40 Hz[41]. Glutamatergic neurons were activated by blue light pulses (5 ms) at 20 Hz, and inhibited using continuous yellow light. Mice with missed injection or cannula locations were excluded. Control mice underwent the same procedures and received the same intensity of laser stimulation.

For pharmacologic experiments, a cannula (RWD Life Science) was implanted above the LPBN (AP, 5.02 mm; ML, ± 1.57 mm; DV, 2.97 mm relative to bregma) for infusion of the GABA$_A$ receptor blocker PTX (0.03 μg kg$^{-1}$). Two stainless steel screws were implanted in the skull for support. The cannula and screws were held in place with dental cement. A stainless steel obturator was inserted into each guide cannula to prevent blockage. Only mice with the correct locations of optical fibers/cannulae and viral expression were used for further analysis.

For prolonged activation of glutamatergic and GABAergic LPBN neurons, DREADD activation was achieved in VgluT2-ires-Cre mice and GAD2-ires-Cre mice where AAV-DIO-hM3Dq-mcherry was injected into the LPBN. Clozapine N-oxide (CNO, Enzo Life Sciences, Inc.) was dissolved in saline to a concentration of 0.5 mg/ml. In Fig. 7, CNO/saline was injected daily over the first week (a total of 7 injections). Consecutive CNO (i.p., 1.5 mg kg$^{-1}$) injections were spaced 24 h apart and mechanical threshold was measured 1 h before each CNO injection (or 23 h after previous CNO injection). No more CNO injection was given after the first week.

## Fiber photometry and endoscopic in vivo calcium imaging

In order to monitor the neuronal activity of glutamatergic LPBN neurons and GABAergic LPBN neurons after AAV-hSyn-DIO-GCaMP7s-WPRE or AAV-EF1α-DIO-eYFP virus were injected into the LPBN of VgluT2-ires-Cre mice and GAD2-ires-Cre mice. After two-three weeks, an optical fiber (230 μm OD, 0.37 NA) was implanted into the LPBN, then each mouse was allowed to recover for 3 weeks before recording. Each mouse was handled for 3 days prior to fiber photometry recording. On day 7, the PWT and Ca$^{2+}$ transient (or eYFP) signal were recorded simultaneously with F-scope-G-2 after the hindpaw was stimulated with pinch. For noxious mechanical stimulation (pinch), an alligator clip (Amazon, Generic Micro Steel Toothless Alligator Test Clips 5AMP) producing 340 g force was applied to the ventral skin surface between the footpad and the heel[82].

A fiber photometry system (Thinker Tech Nanjing Bioscience Inc.) was used for recording[40,83]. At the end of the experiment, all animals were perfused. Only data from animals with correct optical fiber implantation sites and virus expression were included in the analysis. The onset and offset points of stimulation were determined by analyzing videos frame-by-frame or retrieving analog signals with the MatLab R2016a program. After subtracting the noise of the photometry recording system, we smoothed the data with a moving average filter (100 Hz). The change in Ca$^{2+}$ transient value ($\Delta F/F$) from 2 s preceding the onset of hindpaw withdrawal to 4 s after the onset of pinch stimulation were derived by calculating ($F - F0)/F0$ for each pinch stimulation, where F0 is the median Ca$^{2+}$ transient at 2 s preceding pinch stimulation to its onset. The $\Delta F/F$ values of all pinch stimuli were then averaged and plotted with a shaded area indicating the SEM.

In some experiments, in vivo Ca$^{2+}$ imaging with micro-endoscopy was used[45]. Adult mice were anaesthetized with sodium pentobarbital (1%, 40 mg/kg, i.p.) and treated with dexamethasone (0.2 mg/kg, s.c.) to prevent brain swelling and inflammation. During the procedures, a piece of skull (1 mm in diameter) above the LPBN was removed after high-speed dental drilling. AAV-hSyn-DIO-GCaMP7s virus was injected into the LPBN (AP, 5.02 mm; ML, ±1.25 mm; DV, 3.27 mm) in VgluT2-ires-Cre mice. A ProView implant lens probe (~6.1 mm length, 0.5 mm diameter; Inscopix, #1050-002211) was inserted into the brain and targeted to ~50–100 μm above the injection. Three weeks after injection, the baseplate of a miniaturized integrated fluorescence microscope (Inscopix, #1050-002192) was fixed on top of the microendoscope in well-labeled animals. Ca$^{2+}$ was imaged and analyzed as in our previous reports[45].

Three to four days after baseplate surgery, animals were habituated to the attachment of the microscope and environment for 1 day. Ca$^{2+}$ was then imaged in freely-moving mice with von Frey filaments (0.4 g and 1.0 g) or forceps pinch to the

hindpaw. CPN was then performed and Ca$^{2+}$ was imaged -n the same animals 7 days after CPN. Each imaging session included 4 blocks: 20 min habituation and 4 min of free movement followed by 8–12 trials (trial interval, 10 s) of von Frey and pinch stimulation.

Ca$^{2+}$ was imaged in freely-moving mice using the head-attached microscope as in our previous reports[45] (Inscopix; LED power, 0.6–1.0 mW; camera resolution, $1440 \times 1080$ pixels). Mouse behavior was recorded with a video camera (Canon) synchronized with Ca$^{2+}$ imaging using the trigger-out signal from nVista HD[45]. Ca$^{2+}$ imaging videos were analyzed and neuronal signals were detected using Imaging Data Processing (Inscopix) and custom-written scripts in MatLab following published algorithms (PCA-ICA)[45,84,85].

For each neuron, the amplitude of each Ca$^{2+}$ peak was normalized by the standard deviation of the whole Ca$^{2+}$ trace. To visualize the activity patterns of neuron during stimulation, the active event traces of each excited LPBN neuron were aligned by the time when the von Frey filament or pinch was applied to the plantar surface of the left hindpaw. The resulting traces from all LPBN neurons were sorted by their peak activation time during the window and displayed in temporal raster plots. To quantify the difference in activity between pre-CPN and post-CPN conditions, we first calculated the trial-averaged activity of LPBN neurons in response to different sensory stimuli, then compared the responses between pre-CPN and post-CPN conditions within 1 s of stimulation. A significant increase in activity of LPBN neurons in response to von Frey filaments and pinch was found after CPN.

## Open field test

The open field arena was a square box ($50 \times 50$ cm) with opaque Plexiglas walls. Each test mouse was placed in the center of the box and recorded by a camera attached to a computer. The movement was automatically tracked and analyzed by ANY-maze software. The total distance traveled and time spent in the center area ($25 \times 25$ cm) were measured. To assess the effect of optogenetic activation or suppression of LPBN neurons on locomotor activity, mice were tested for a 15-min session consisting of 5-min Pre, 5-min Light, and 5-min Post periods. Blue (473 nm, 5 ms, 5–10 mW) or yellow (589 nm, 1 Hz, 999 ms, 10 mW) laser light was delivered bilaterally during the light On phase. The box was cleaned with 70% ethanol after each trial. The immobility time and time spent in the center area were automatically analyzed by ANY-maze software 5.3.

## Real-time place avoidance (RTPA) test

Mice were placed in a custom-made $50 \times 27 \times 30$ cm two-chamber apparatus made with opaque acrylic Plexiglas that had distinct stripe patterns. Each mouse was placed in the center and allowed to explore both chambers without light stimulation for 10 min. After exploration, the mouse indicated a small preference for one of the two chambers. Subsequently, blue light stimulation (20 Hz, 5-ms pulse width, ~5 mW for CaMKIIα-positive neurons and VgluT2-positive neurons) was delivered whenever the mouse entered or stayed in the preferred chamber, and the light was turned off when the mouse moved to the other chamber (stimulation phase) for 10 min. Finally, the mouse was allowed to freely explore both chambers without blue light stimulation for 10 min. The RTPA location plots and total time on the stimulated side were recorded and counted with the ANY-maze software 5.3 via a webcam (Logitech web-camera)[41,42].

## Real-time place preference (RTPP) test

Mice were placed in a custom-made $50 \times 27 \times 30$ cm two-chamber apparatus made with opaque acrylic Plexiglas that had distinct stripe patterns. Each mouse was placed in the center of the apparatus and allowed to explore both chambers without light stimulation for 10 min. After exploration, the mouse normally showed a small preference for one of the two sides. Subsequently, blue light stimulation (40 Hz, 5-ms pulse width, ~5 mW) was delivered whenever the mouse entered or stayed in the non-preferred chamber, and the light was turned off when the mouse moved to the other chamber (stimulation phase) for 10 min. Finally, the mouse was allowed to freely explore both chambers without blue light stimulation for 10 min. The RTPP location plots and total time on the stimulated side were recorded and counted with the ANY-maze software 5.3 via a webcam (Logitech web-camera)[20,41].

## Conditioned place aversion (CPA) test

To validate the activation of glutamatergic neurons with (DREADD-hM3Dq), AAV-hSyn-DIO-hM3Dq-mCherry (80 nl/side) and AAV-EF1α-DIO-mCherry virus (80 nl/side) were injected into VgluT2-ires-Cre mice. Three weeks later, the mice were subjected to a 2-chamber classical conditioned place aversion (CPA) test. Each mouse was first habituated to the chamber on days 1 and 2 (baseline) for 30 min each day. On day 3, each mouse was placed in the center of the box and allowed to explore both chambers without stimulation (screening test) for 10 min. Generally, after exploration, the mouse showed a small preference for one of the chambers. On days 4–6, each mouse was confined to the preferred chamber with saline injection (i.p.) for 30 min in the morning, and 4 h later confined to the non-preferred chamber with CNO injection (1.5 mg/kg, i.p.) for 30 min in the afternoon. On the final day (day 7), each mouse was allowed to explore both chambers with saline or CNO for 10 min, and its behavior was recorded and counted with the ANY-maze software 5.3 via a webcam (Logitech web-camera).

**Immunohistochemical staining**. Staining with c-Fos was used to assess the effect of in vivo optogenetic and pharmacogenetic manipulations on neuronal activity in the LPBN. Mice were stimulated with blue or yellow light for 10 min. They were then transcardially perfused with 0.1 M PBS and 4% paraformaldehyde in PBS about 60–90 min after light stimulation. The brain was then removed and fixed in 4% paraformaldehyde buffer at 4 °C overnight. After fixation, the brain was cryoprotected in 30% sucrose (wt/vol) at 4 °C. Free-floating sections (35–40 µm) cut on a cryostat (Leica CM 1950) were used for immunohistochemical staining, except for that implanted with optical fibers or cannulae, which were mounted on slides immediately after sectioning. After rinsing with ice-cold methanol (10 min) and blocking with 10% (wt/vol) normal bovine serum for 1 h at room temperature, the sections were incubated (12–24 h at 4 °C) with the primary antibodies anti-c-Fos (1:500, Guinea pig, SYSY), anti-c-Fos (1:1000, rabbit, SYSY), anti-CaMKIIα (1:1000, rabbit, Abcam), and anti-vGAT (1:800, mouse, SYSY). After rinsing, the sections were incubated with fluorophore-conjugated secondary antibodies for 2 h at room temperature (1:1000, anti-Guinea pig Alexa Fluor 488, anti-rabbit Cy3, anti-Guinea pig Cy3, anti-rabbit Alexa Fluor 488, Jackson; 1:1000, anti-mouse Alexa Fluor 488, Invitrogen). The antibodies were diluted in phosphate-buffered saline containing 5% BSA. For c-Fos quantification, mice were exposed to blue (473 nm, 20 Hz/40 Hz, 5-ms pulse width, 5 mW) or yellow (589 nm, 1 Hz, 999-ms pulse width, 10 mW) laser for 10 min, perfused for 60 min, and sections were cut. After anti-Fos immunohistochemistry, nuclei were stained with DAPI, and confocal images were captured on an Olympus FV-1200 microscope. Cell counting was carried out manually using Image J-win64.

**RNAscope in situ hybridization**. RNAscope Fluorescent Multiplex Assays with the GAD1 probe were used to determine the overlap of different neuron types in the LPBN. Brain sections from VgluT2-ires-Cre mice injected with rabies virus (RV-EnvA-DsRed) in the LPBN were cut at 40 µm on a cryostat (Leica CM 1950), and collected into DepC-PBS. Floating sections were mounted on slides, heated at 60 °C for 2 h and kept at –80 °C before experiments. Slides were treated thrice with ethanol (5 min each time) and air-dried at room temperature. Then, the slides were pretreated with hydrogen peroxide at room temperature for 10 min and washed twice in DepC-PBS (2 min each time). Protease digestion was performed in a 40 °C HybEZ oven for 15 min. After washing in DepC-PBS for 3 min and rinsing in DepC-ddH$_2$O, slides were hybridized with pre-warmed GAD1 probe, Slc17a6 probe (VgluT2), positive control probe (peptidylprolyl isomerase B, Ppib), or negative control probe [dihydrodipicolinate reductase (dapB) gene] in the 40 °C HybEZ oven for 2 h. The signal amplification fluorescent label was TSA-based. Anti-DsRed staining was performed after the RNAscope staining to define DsRed$^+$ neurons. RNAscope multiplex fluorescent reagent kit v2 and appropriately designed probes (ACDBio Inc.) were used. The probes were alternated across all sections to ensure that one posterior section and one anterior section from each region was analyzed with each probe type. Poorly stained slides were not analyzed.

**Image acquisition and quantification**. Samples were imaged on a Olympus FV-1200 laser scanning confocal microscope. All immunohistochemical staining samples were imaged at ×10 resolution. In situ samples were imaged at ×20 resolution at three z-positions. All z-positions for each slice were merged into a single image in Adobe Photoshop CC for quantification. The captured neurons and Fos-expressing neurons in all immunohistochemical and in situ hybridization experiments were manually counted, and percentages were calculated for each animal before averaging the percentages across animals. To evaluate the relative marker expression (Fig. 1a–h and Fig. 7g–j), we summed the cell counts across brain sections from 5 or 6 mice.

**Electrophysiological recording in acute brain slices**. For whole-cell recordings, each mouse was anesthetized with isoflurane and the brain dissected rapidly and immersed in ice-cold oxygenated (95% O$_2$/5% CO$_2$) artificial cerebrospinal fluid (ACSF) (in mM: 124 NaCl, 2 KCl, 2 MgSO$_4$, 1.25 KH$_2$PO$_4$, 2 CaCl$_2$, 26 NaHCO$_3$, 10 D-glucose, pH 7.4, 300 mOsm). Coronal slices (300 µm) containing the LPBN complex were cut on a vibratome (Microm HM 650 V), and then allowed to recover in ACSF. Slices were placed on glass coverslips and submerged in a recording chamber. Whole-cell voltage-clamp recordings were amplified with Multiclamp 700B (filtered at 2 kHz), digitized with Digidata 1440 A (5 kHz), and recorded using pClamp 10.4 software (RRID: SCR_011323). Borosilicate glass pipettes (Sutter instruments) (3–5 MΩ) were pulled on a horizontal pipette puller (P97, Sutter Instruments) and filled with artificial intracellular fluid (in mM: 130 CsCl, 1 EGTA, 0.5 CaCl$_2$, 2 MgCl$_2$, 2 QX-314, 2 ATP-Na, 0.4 GTP-Na; pH 7.3, 285–290 mOsm). Both mCherry$^+$ (glutamatergic) and eYFP$^+$ (GABAergic) LPBN neurons were visualized under the Olympus microscope. Neurons, axons, and synaptic terminals expressing ChR2 were activated by 10-ms flashes of blue light (473 nm, 1 Hz) from a single-wavelength laser system (Newdoon Inc., Hangzhou, China). The blue light was focused on the aperture of the microscope objective, producing a wide-field exposure around the recorded cell. To confirm that post-synaptic currents were monosynaptic, eIPSP signals were recorded at –70 mV in ACSF containing tetrodotoxin (TTX; 1 µM), followed by a combination of TTX and 4-aminopyridine (4-AP; 100 µM), and finally by a combination of TTX, 4-AP, and bicuculline (10 µM).

To establish the efficiency of ChR2 expression, blue laser light (473 nm, 1, 5, and 20 Hz) was delivered to a neuron expressing ChR2 and the evoked time-locked membrane depolarization and action potentials were recorded. Whole-cell patch-clamp recordings were made at room temperature. The internal solution contained (in mM): 75 CsCl, 75 CsF, 2.5 MgCl$_2$, 5 HEPES, 2.5 EGTA, 14 creatine phosphate-Tris, 4 Mg-ATP, 0.3 GTP-Tris. Blue laser light (473 nm, 1, 5, and 20 Hz) was delivered to the recorded neuron and used to activate glutamatergic neurons (eYFP) in the LPBN. Signals were filtered at 2 kHz, sampled at 10 kHz, and analyzed using Igor pro software.

In some experiments, current-clamp recordings were performed using glass electrodes (3–5 MΩ) containing (in mM) 135 K-gluconate, 6 NaCl, 10 HEPES, 0.5 EGTA, 10 Na$_2$-phosphocreatine, 4 Mg-ATP, 0.3 Na2-GTP (pH 7.2 adjusted with KOH; 290–300 mOsm).

All electrophysiological data were analyzed off-line using the Neuromatic package (Think Random) in Igor Pro software (WaveMetrics). Off-line analysis was performed by averaging five traces. Light-evoked IPSC peak amplitude, onset latency, time to peak, rise time, and decay time were analyzed. The onset latency of the light-evoked IPSCs was defined as the time from the onset of the stimulus to the first measurable deflection of the potential from baseline. After recording, the slices were fixed with 4% paraformaldehyde in PBS for immunohistochemical staining.

**Statistical analysis**. All experiments and data analyses were conducted blinded, including the immunohistochemistry, electrophysiology and behavioral analyses. Data were analyzed with Graphpad Prism v.6.0, Image J1.48V, Image J-win64, Olympus FV10-ASW 4.0a Viewer, Adobe Illustrator CS6 (version 16), Adobe Photoshop CC 2017, Microsoft office 2013, Adobe Acrobat Pro 2017, IBM SPSS_26.0, and MATLAB R2016a software.

Since our data were continuous variables, parametric statistics were used. The homogeneity of variance was assessed by Levene's test of Equality of Error Variances. Square-root transformation or logarithmic transformation was performed for data that did not meet the homogeneity of variance test. Behavioral data were analyzed by one-way or two-way analysis of variance (ANOVA) followed by Bonferroni's test for multiple comparisons, and the unpaired t-test for two-group comparisons. Imaging of calcium activity data were analyzed by Wilcoxon Signed Rank Test. For immunofluorescence analysis, data were analyzed using one-way ANOVA or unpaired Student's t-tests. Statistical significance was accepted at the level of $P < 0.05$ with asterisks in figures denoting $P$ values as follows: $*P < 0.05$, $**P < 0.01$, $***P < 0.001$, $****P < 0.0001$. All data are presented as the mean ± standard error of the mean (SEM). No statistical methods were used to predetermine sample sizes, but our sample sizes are similar to those reported in previous publications[41,43]. Supplementary Table 1 provided statistical data for all figures.

**Reproducibility**. Experiments were repeated independently with similar results at least three times. Micrographic images presented in figures are representative ones from experiments repeated independently: Fig. 1a, b (four times), Fig. 1e (three times), Fig. 1g (three times), Fig. 6a (five times), Fig. 7g (four times), Fig. 8b (three times), Fig. 8d (four times), Fig. 9j (three times).

**Reporting summary**. Further information on research design is available in the Nature Research Reporting Summary linked to this article.

## Data availability

Source data are provided with this paper. All data are contained in the main text and the supplementary materials, and are available from the corresponding author upon reasonable request. The source data underlying Figs. 1c, d, f, h, k, n; 2d, e, h, j; 3b–e, h; 4b, c, f, g, j–m; 5b, c, e, f, h–k; 6b, c, e, g, i–l, n; 7b, c, f, h, i, j; 8f, g, j, k; 9b, e, g, i, k and Supplementary Figs. 9c; 2f, g; 3b, d; 5b, c, e and 6b, e are provided as a Source Data file. Allen Brain Atlas (http://www.brain-map.org).

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

## Acknowledgements

We thank Zhen-zhong Xu, and I.C. Bruce for discussions and reading the manuscript. We thank Yu-dong Zhou, Kun Chen, Hong-bin Yang, Hui-Fang Lou, and Li-Ya Zhu for technical support. This work was supported by the National Natural Science Foundation of China (31800880, 81527901, and 81821091), the National Key Research and Development Program (2016YFA0501000 and 2016YFC1306700), Key Realm R&D Program of Guangdong Province (2019B030335001), the Science and Technology Planning Project of Guangdong Province (2018B030331001), CAMS Innovation Fund for Medical Sciences (2018PT31041; 2019-I2M-5-057), and Fundamental Research Funds for the Central Universities (2019FZA7009).

## Author contributions

Conceptualization, L.S. and S.D.; Methodology, L.S., R.L., Z.G.Z., X.Y.L., and S.D.; Investigation, L.S., R.L., F.G., M.Q.W., X.L.M., K.Y.L., H.S., C.L.X., Y.Y.L.; X.J.L.; Visualization, L.S.; Writing, L.S. and S.D.; Funding Acquisition, L.S. and S.D.; Resources, L.S. and S.D.; Supervision, L.S., Y.Q.Y., Z.C., X.Y.L., and S.D.; Statistical analysis, L.S., M.Y.W.

## Competing interests

The authors declare no competing interests.
