## [Peer Review File · Nature Communications]

Reviewers' comments:

Reviewer #1 (Remarks to the Author):

This study investigated the contribution of the lateral parabrachial nucleus (LPBN) to the development of neuropathic pain in a mouse model of neuropathic pain, induced by ligation of common peroneal nerve (CPN). The authors showed several interesting findings using optogenetic and chemogenetic approaches. They demonstrated distinct roles of excitatory and inhibitory neurons of the LPBN for pain regulation. They found that LPBN is not only important for relaying physiological pain and but also critical for the processing of neuropathic pain. They found a relatively small population of GABAergic neurons in the LPBN (10% of total neurons), but this population of inhibitory neurons plays a gating role in the development of neuropathic pain. Overall, this is an elegant study, and the data are clearly presented and look convincing. Both gain-of-function and loss-of-function approaches were employed to support the conclusions of this study. Neuropathic pain is also carefully assessed. In addition to hyperalgesia, real-time place preference (RTPP) test and CPA were assessed for aversive/emotional aspect of pain. There are also several concerns that should be addressed before the paper is accepted for publication.

An important finding of this study is to demonstrate a gating role of GABAergic neurons during the development of neuropathic pain. There are only 10% GABAergic neurons in the LPBN, and this population is much smaller than that in the spinal cord where gate control theory is based. It will be very interesting to demonstrate whether inhibitory synaptic transmission (IPSC) in the LPBN is diminished after nerve injury, as shown in the spinal cord.

Disinhibition in the spinal cord pain circuit is induced by the activation of glial cells which can produce BDNF and pro-inflammatory cytokines (PMID: 16355225; PMID: 18480275). It is unclear if neuroimmune regulation also plays a role in the LPBN in neuropathic pain.

Fig. 6 is impressive to show that prolonged activation of GABAergic LPBN neurons can prevent the development of neuropathic pain. It is also important to know if activation of these inhibitory neurons can reverse neuropathic pain in the late-phase (after 3 weeks), especially from the therapeutic perspective for the treatment of neuropathic pain. This issue could be addressed by additional discussion. It is known that late-phase neuropathic pain could be regulated by different mechanisms.

Line 542: Only male mice 8–10 weeks of age (20–35 g) were used. Please discuss if there is sex difference in the investigated mechanisms. Sex dimorphism has been reported in several pain mechanisms.

How many mice in total were used for this study?

Reviewer #2 (Remarks to the Author):

This manuscript by Sun and co-workers investigates the role and the underlying mechanisms of the lateral parabrachial nucleus (LPBN) in processing neuropathic pain. By using multiple complementary approaches, the authors report that the activity of GABAergic LPBN neurons, a minor neuronal population that directly innervates glutamatergic LPBN neurons, plays a gating role in the sensitization of glutamatergic neurons, which is necessary and sufficient for the development and transmission of neuropathic pain. These results led the authors to suggest that a delicate balance between excitatory and inhibitory neuronal activity in the LPBN governs the development and maintenance of neuropathic pain. Overall the studies are relatively complete and compelling. The following specific comments should be considered by the authors for the overall improvement of the manuscript.

1) It is strange that optogenetic activation of LPBNVgluT2 neurons expressing ChR2-eYFP induced marked mechanical allodynia and thermal hyperalgesia in sham-operated mice but not in CPN-ligated mice (page 8, lines 147-153; Fig. 2d, e), whereas optogenetic silencing of LPBNCaMKII α neurons increased the PWT and PWL in both sham-operated and CPN-ligated mice (page 10, lines 189-193; Fig. 3b, c), why optogenetic activation of LPBNVgluT2 neurons could not induce pain hypersensitivity in CPN-ligated mice? Will optogenetic silencing of LPBNVgluT2 neurons produce the same results as optogenetic silencing of LPBNCaMKII α neurons?

2) Pharmacogenetic activation of GABAergic LPBN neurons alleviates the mechanical allodynia of CPN-ligated mice (page 13, lines 254-267; Fig. 4m, n), will it have any effect on aversive emotional responses of CPN-ligated mice?

3) Optogenetic activation of GABAergic LPBN neurons alleviates neuropathic but not physiological pain, whereas optogenetic inactivation of GABAergic LPBN neurons is sufficient to induce pain hyperalgesia in naïve mice. Whether loss of GABAergic LPBN neurons occurs in CPN-ligated mice?

4) To validate the monosynaptic innervation of glutamatergic LPBN neurons by local GABAergic neurons, additional evidence should be provided to determine whether optogenetic activation or silencing of GABAergic LPBN neurons could respectively inhibit or enhance the activity of glutamatergic LPBN neurons?

5) Which subtype(s) of GABAergic LPBN neurons is involved in gating glutamatergic LPBN neurons?

6) In page 14, lines 279-281, the meaning of the sentence "...we found that light inactivation of GABAergic LPBN neurons drastically decreased both mechanical (Fig. 5b) and thermal (Fig. 5c) hyperalgesia in naïve (without CPN ligation) GAD2-ires-Cre mice transfected with GtACR1-eYFP..." is not match with the results shown in the fig. 5b, c.

Reviewer #3 (Remarks to the Author):

GENERAL COMMENTS

The paper described an impressive body of work, supporting the conclusion that the lateral parabrachial nucleus (LPB) is involved in chronic pain. Reducing enthusiasm for this manuscript are several issues with experimental design and data interpretation, and deficits in the scientific premise, reflecting a failure to adequately consider relevant, existing knowledge on the role of LPB in pain, and a failure to relate the current findings to this knowledge.

Most of the data describe behavioral experiments that largely replicate previous findings, a worthwhile goal. Other data describe immunohistological, calcium imaging and electrophysiological experiments, some of which raise issues.

SPECIFIC COMMENTS

INTRODUCTION

The introduction fails to reflect current knowledge of the pathophysiology of neuropathic pain, and some of it has little to do with the topic of this study. At places it reads like a series of unrelated statements about the pain literature, statements that do not relate to the overarching hypothesis or goal of this study. What relevant discussion about LPB circuits exists fails to reflect the rich literature on this structure and its role in pain. It is simply wrong to state that "it is not clear whether and how PBN" is involved in neuropathic pain. See, for example, PMID 8985905,

28833700, 29703377, 29862375, 31479066, 32217613.

RESULTS

Figure 1

Here, and elsewhere, the authors use “noxious pinch” to activate LPB neurons, and to compare activation patterns in animals with chronic pain with those in controls. For these analyses to be valid, a noxious stimulus that produces reproducible responses must be used. There is no indication that this was done.

The authors argue that CPN injury model results in specific activation of CamKIIa neurons. Yet, they do not appear to explicitly test this assertion. It seems that they need to directly compare, between sham and CPN conditions, the proportion of CamKIIa and GAD67 expressing cFos. The analyses reported here do not support the authors’ conclusion.

Here, and in other immunohistochemical (and RNA scope) experiments, the absence of controls (e.g., omission or adsorption of primary antibody) reduces the rigor of these experiments. Also reducing rigor is the absence of information on sampling procedures, the number of sections counted, and quantification procedures.

Contrary to the stated assumption, DAPI is not a neuronal marker, and should not be used as a proxy for one.

It is inaccurate to state that “glutamatergic LPB neurons are selectively activated in the mechanical allodynia”. What the authors show is that glutamatergic neurons in LPB are more active in CPN injured animals in response to a single hindpaw pinch stimulus. This is not allodynia.

Figure 2

Feasibility experiments for optogenetic stimulation show that opsin-containing neurons can entrain stimuli at 5Hz stimuli. Behavioral experiments use stimuli of 20Hz, without ascertaining that LPB neurons can entrain such stimuli.

The finding that tests of anxiety were not affected by the induction of chronic pain are surprising, and contradict literature indicating otherwise. This should be considered and discussed.

The authors state that optogenetically activating glutamatergic neurons causes hyperalgesia; that is inaccurate. They present evidence for allodynia and aversiveness, but not for hyperalgesia.

The statement that activating glutamatergic neurons causes signs of pain is not “striking”, as the authors state. There exists substantial prior research indicating that activating excitatory neurons in LPB causes pain-like behaviors, as well as pronounced aversion.

Figure 4

There is no explicit consideration of the likelihood that optogenetic stimulation resulted in antidromic activation of neurons, and their axon collaterals, in other nuclei. This is a potential confound to both the experiments using the vGAT-hChR2 mice and those using the GAD2-Cre mouse with local injection of AAV-hChR2s. Therefore, it is wrong to assume that optical stimuli exclusively activate neurons intrinsic to LPB. Similarly, the assumption that local infusion of picrotoxin selectively inhibits somata needs to be tempered by knowledge that picrotoxin is also a channel blocker, and can affect activity of afferents extrinsic to LPB.

The paper might benefit from referencing important papers on the role of GABAergic inputs from

CeA to LPB. For example, PMID 31577943, 26733798, 32217613.

Figure 5

The stated goal of the experiments described here is to test the hypothesis that GABAergic LPB neurons "gate neuropathic pain". The relevant experiment here would have been to compare data from sham and CPN conditions. This experiment was not performed.

The comparisons of labeled neurons is also not consistent with the relevant statistical hypothesis. A meaningful comparison here would be to compare the percentage of c-fos positive GAD2 neurons in GtACR1 versus EYFP, and to separately compare the percentage of c-Fos positive CamkIIa neurons in GtACR1 versus EYFP.

In patch-clamp experiments it is surprising that large IPSCs were evoked while holding neurons near the chloride reversal potential. It might be instructive to discuss this finding.

Data analysis

- There is no justification for the use of parametric statistics.
- No a priori sample size estimation was performed. Stating that "sample sizes are similar to those reported in previous publications" does not reduce this deficiency.
- Several important figures fail to depict individual data points.
- Both Bonferroni and Tukey post-hoc corrections are used; why?

DISCUSSION

It would be appropriate to revise the discussion after the analyses suggested above are implemented.

It would be useful and appropriate to consider the fact that LPB interacts with a large number of pain-related brain regions. This should temper the strong conclusion that manipulating PB neurons is both necessary and sufficient to induce neuropathic pain.

Parabrachial nucleus circuit governs the development and transmission of neuropathic pain

Summary of main new results and major changes

We thank the reviewers for their constructive comments and suggestions. We have conducted substantial new experiments and extensively revised our manuscript. The new results and re-organized data are presented as Fig.1c, d, g, h; Fig. 2b; Fig. 4; Fig. 6h, i, r–u; Supplemental Fig. 2; and Supplemental Figs 4–6 in the revised manuscript. Some new data are presented in the ‘point-by-point reply to reviewers’. The main new results and major changes in the revised manuscript are summarized below:

1. We further studied the noxious and mechanical stimulation-induced Ca^{2+} signaling in glutamatergic LPBN neurons using a head-attached microscope for *in vivo* Ca^{2+} imaging; this has a higher sensitivity and spatial resolution than fiber photometry so that both noxious pinch and Von Frey mechanical stimulation could be used to induce detectable Ca^{2+} activity changes in LPBN neurons. We found that CPN ligation enhanced the activity of glutamatergic LPBN neurons induced by either noxious pinch or Von Frey mechanical stimuli, strengthening our previous results with fiber photometry. These results are summarized as **Supplemental Fig. 2** in the revised manuscript.
2. Using patch-clamp recordings in brain slice preparations, we showed that action potential firing of the LPBN neurons nicely followed the 20-Hz optogenetic stimulation, validating the physiological relevance of the light stimulation frequency used for behavioral experiments. This result is presented in **Fig. 2b** in the revised manuscript.
3. We showed that optogenetic silencing of LPBN VgluT2 neurons elevated the threshold of pain sensation induced by mechanical or thermal stimulation in both control and CPN-ligated mice. Furthermore, optogenetic silencing of LPBN

VgluT2 neurons in CPN-ligated mice also induced real-time place-preference behavior, which is associated with the seeking of pain relief. These results are similar to our previous data from experiments on the optogenetic silencing of LPBN CaMKII α neurons, consistent with the finding that CaMKII α neurons are extensively overlapped with VgluT2 neurons in the LPBN. These results are presented as **Fig. 4** in the revised manuscript.

4. We showed in electrophysiological recordings from brain slices that a brief light stimulation (5 s) of GABAergic LPBN neurons expressing ChR2 significantly inhibited the firing frequency of glutamatergic LPBN neurons for >10 s, accompanied by membrane hyperpolarization. These results, presented as **Fig. 6r–u** in the revised manuscript, provide direct evidence of functional inhibition of glutamatergic LPBN neuronal activity mediated by local GABAergic neurons.
5. We found that CPN ligation did not change the density of GABAergic LPBN neurons. However, the release probability of these neurons was decreased after CPN ligation, as evidenced by an increased paired-pulse ratio of evoked IPSCs. These results indicate that the function of synaptic transmission, rather than the number of GABAergic LPBN neurons, is affected by neuropathic pain. These results are presented as **Supplemental Fig. 6** in the revised manuscript.
6. We showed that pharmacogenetic activation of GABAergic LPBN neurons in CPN-ligated mice also induced conditioned place-preference behavior, indicating relief of the aversive emotion accompanying neuropathic pain. Pharmacogenetic activation of GABAergic LPBN neurons was confirmed by immunostaining of c-Fos expression in these neurons. These results are summarized as **Supplementary Fig. 5** in the revised manuscript.
7. In the previous version of the manuscript, we used either Tukey's or Bonferroni's test for statistical analysis. In the revised manuscript, we used Bonferroni's test

for the unified statistical analysis of all the data presented. Although the P values of the data analyzed with Bonferroni's test are slightly different from those with Tukey's test, the conclusions of statistical significance are same using either test. For the convenience of comparison, the detailed P values of the data analyzed with Tukey (previous version of the manuscript) and Bonferroni's test (revised manuscript) are presented in Table R2 in the "point-by-point response to reviewers' comments" at the item of the reply to Reviewer 3.

8. Additional new results and changes in the revised manuscript are described in detail in the following "point-by-point response to reviewers' comments". The changed text in the revised manuscript is marked by red letters

Point-by-point response to reviewers' comments

(Reviewers' comments are in italics)

Reviewer #1 (Remarks to the Author):

An important finding of this study is to demonstrate a gating role of GABAergic neurons during the development of neuropathic pain. There are only 10% GABAergic neurons in the LPBN, and this population is much smaller than that in the spinal cord where gate control theory is based. It will be very interesting to demonstrate whether inhibitory synaptic transmission (IPSC) in the LPBN is diminished after nerve injury, as shown in the spinal cord.

We thank the reviewer for the positive comments. Following the reviewer's suggestion, we examined the paired pulse ratio (PPR) of IPSCs evoked in the LPBN glutamatergic neurons by optogenetic stimulation of local GABAergic neurons, and found a significant increase in CPN-ligated mice, indicating a decreased release probability from LPBN GABAergic neurons. On the other hand, the density of

GABAergic neurons did not change after CPN ligation. Nevertheless, it is possible that diminished inhibitory synaptic transmission in GABAergic LPBN neurons may at least in part contribute to the development of neuropathic pain. These results are summarized as **Supplemental Fig. 6** and discussed in the revised manuscript.

Disinhibition in the spinal cord pain circuit is induced by the activation of glial cells which can produce BDNF and pro-inflammatory cytokines (PMID: 16355225; PMID: 18480275). It is unclear if neuroimmune regulation also plays a role in the LPBN in neuropathic pain.

We agree that it will be interesting to explore in future whether glial cells and neuroimmune regulation play a role in the LPBN in neuropathic pain. We have discussed this issue briefly in the revised manuscript.

Fig. 6 is impressive to show that prolonged activation of GABAergic LPBN neurons can prevent the development of neuropathic pain. It is also important to know if activation of these inhibitory neurons can reverse neuropathic pain in the late-phase (after 3 weeks), especially from the therapeutic perspective for the treatment of neuropathic pain. This issue could be addressed by additional discussion. It is known that late-phase neuropathic pain could be regulated by different mechanisms.

We thank the reviewer for raising this important issue. Following the reviewer's suggestion, we have discussed this issue in the revised manuscript.

Line 542: Only male mice 8–10 weeks of age (20–35 g) were used. Please discuss if there is sex difference in the investigated mechanisms. Sex dimorphism has been reported in several pain mechanisms.

We agree that the role of sex dimorphism in the development of neuropathic pain in

LPBN is an interesting topic and worth further study in future. We have cited some references reporting sex dimorphism in chronic pain and briefly discussed this issue in the revised manuscript.

How many mice in total were used for this study?

We used 420 mice in total in our study.

Reviewer #2 (Remarks to the Author):

1) It is strange that optogenetic activation of LPBN^{VgluT2} neurons expressing ChR2-eYFP induced marked mechanical allodynia and thermal hyperalgesia in sham-operated mice but not in CPN-ligated mice (page 8, lines 147-153; Fig. 2d, e), whereas optogenetic silence of LPBNCaMKII α neurons increased the PWT and PWL in both sham-operated and CPN-ligated mice (page 10, lines 189-193; Fig. 3b, c), why optogenetic activation of LPBN^{VgluT2} neurons could not induce pain hypersensitivity in CPN-ligated mice? Will optogenetic silence of LPBN^{VgluT2} neurons produce the same results as optogenetic silence of LPBNCaMKII α neurons?

The reason that optogenetic activation of LPBN^{VgluT2} neurons did not further enhance pain hypersensitivity in CPN-ligated mice may be that the ligation already induced hypersensitivity to the maximum extent, which would mask the effects of optogenetic activation of LPBN^{VgluT2} neurons. The results that the effect of optogenetic activation of LPBN^{VgluT2} neurons did not add to that of the CPN ligation on pain sensation suggest that the two types of treatment share the same mechanisms for inducing pain hypersensitivity. We have discussed this issue briefly in the revised manuscript.

Following the reviewer's suggestion, we conducted new experiments and found that optogenetic silencing of LPBN VgluT2 neurons had the same effect as that

induced by optogenetic silencing of LPBN CaMKII α neurons, consistent with the finding that CaMKII α neurons are extensively overlapped with VgluT2 neurons in the LPBN. These results are presented as **Fig. 4** in the revised manuscript.

2) Pharmacogenetic activation of GABAergic LPBN neurons alleviates the mechanical allodynia of CPN-ligated mice (page 13, lines 254-267; Fig. 4m, n), will it have any effect on aversive emotional responses of CPN-ligated mice?

We performed conditioned place preference (CPP) and found that pharmacogenetic activation of GABAergic LPBN neurons significantly increased the time the CPN-ligated mice stayed in the stimulated side coupled with CNO injection, consistent with the relief of aversive emotion induced by the activation of GABAergic LPBN neurons. These results are summarized as **Supplemental Fig. 5** in the revised manuscript.

3) Optogenetic activation of GABAergic LPBN neurons alleviates neuropathic but not physiological pain, whereas optogenetic inactivation of GABAergic LPBN neurons is sufficient to induce pain hyperalgesia in naïve mice. Whether loss of GABAergic LPBN neurons occurs in CPN-ligated mice?

We assessed the density of GABAergic neurons in the LPBN and did not find a significant difference between control and CPN-ligated mice. However, we found a significantly increased paired pulse ratio (PPR) of IPSCs recorded in the glutamatergic LPBN neurons evoked by optogenetic stimulation of local GABAergic neurons, suggesting a decreased release probability from GABAergic LPBN neurons. It is possible that diminished inhibitory synaptic transmission in GABAergic LPBN neurons may at least in part contribute to the development of neuropathic pain. These results are summarized as **Supplemental Fig. 6** and briefly discussed in the revised manuscript.

4) To validate the monosynaptic innervation of glutamatergic LPBN neurons by local GABAergic neurons, additional evidence should be provided to determine whether optogenetic activation or silence of GABAergic LPBN neurons could respectively inhibit or enhance the activity of glutamatergic LPBN neurons?

We appreciate the reviewer's suggestion. We found that optogenetic activation of GABAergic LPBN neurons significantly decreased the firing rates of glutamatergic LPBN neurons. These results are presented as **Fig. 6r–u** in the revised manuscript.

5) Which subtype(s) of GABAergic LPBN neurons is involved in gating glutamatergic LPBN neurons?

To determine the subtypes of GABAergic LPBN neurons, we did double *in situ* hybridization in the LPBN to examine the mRNA co-expression of GAD1 with CCK, PV, or SST, markers of the three major subtypes of GABAergic neurons in the brain. We found that the ratios of CCK, PV, or SST mRNA-positive cells in GAD1 mRNA-expressing cells were 2.9%, 13.8%, and 36.2% respectively in the LPBN. We noted distinct subtype distribution patterns in PBdl and PBdel. The ratios of CCK, PV, and SST mRNA-positive cells in GAD1 mRNA-expressing cells were 3.4%, 3.7%, and 89.0% respectively in PBdl, whereas these ratios were 2.4%, 44.7%, and 4.7% in PBel (Fig. R1a–e). However, we found that the expression of these markers was not limited to GABAergic LPBN neurons. For example, the GAD1 mRNA-positive neurons only constituted 2.1%, 7.5%, and 21.3% of CCK, PV, and SST mRNA-expressing cells, respectively, in PBdl, 15.5%, 68.4%, and 54.0% in PBel, and 3.2%, 24.5%, and 29.4% in the whole LPBN (Fig. R1a–g). Although we did not systemically examine the expression of the three markers in glutamatergic neurons, we found that the ratio of SST mRNA-expressing cells among Vglut2 neurons was 23.2%, 7.3%, and 17.5% in PBdl, PBel, and the whole LPBN, respectively, whereas the ratio of Vglut2-expressing neurons among SST-positive cells was 75.2%, 33.3%, and 66.5%

in PBdl, PBel, and the whole LBP (Fig. R1h and i). These findings prevented us from using the selective optogenetic or pharmacogenetic manipulation of activity in these subtypes of GABAergic LBP neurons. Further efforts are needed in future to find suitable approaches to identifying the specific subtype of GABAergic LBP neurons involved in gating glutamatergic LBP neurons.

Fig. R1. Distribution of CCK, PV, and SST among glutamatergic and GABAergic neurons in the LPBN.

(a–c) Representative images of *in situ* hybridization showing co-localization (white arrowheads) of CCK (upper), PV (middle), and SST (lower, green) with GAD1 (red). Panels in the 5th column are magnified views of the rectangular regions of panels in the 4th column. Scale bars, 200 μm for panels in the 1st–4th columns and 100 μm for panels in the 5th column.

d. Summarized proportions of CCK, PV, and SST-positive cells among GAD1-positive neurons in the whole LPBN.

e. Proportions of GAD1-positive neurons among CCK, PV, and SST-positive cells in the whole LPBN.

f. Proportions of CCK, PV, and SST-positive cells among GAD1-positive neurons in PBdl and PBel.

g. Proportions of GAD1-positive neurons among CCK, PV, and SST-positive cells in PBdl and PBel.

- h. Representative images of *in situ* hybridization showing co-localization (white arrowheads) of SST (red) with GAD1 (green) and SLC17a6 (Vglut2, blue) in PBdl (2nd and 3rd panels from left) and PBel (4th and 5th panels from left). Scale bars, 200 μ m for the 1st panel from left and 100 μ m for the right 4 panels.
- i. Proportions of SST-positive cells among Vglut2 neurons (left panel) and proportions of SLC17a6 (Vglut2)-positive neurons among SST-positive cells (right panel) in PBdl, PBel, and the whole LPBN.

6) In page 14, lines 279-281, the mean of the sentence “...we found that light inactivation of GABAergic LPBN neurons drastically decreased both mechanical (Fig. 5b) and thermal (Fig. 5c) hyperalgesia in naïve (without CPN ligation) GAD2-ires-Cre mice transfected with GtACR1-eYFP...” is not match with the results shown in the fig. 5b, c.(Fig. 6b,c in the revised manuscript)

We are very sorry for the mistake in the description. We have re-phrased the sentence as “light inactivation of GABAergic LPBN neurons drastically decreased the threshold and latency of the paw-withdrawal responses evoked by mechanical (Fig. 6b) and thermal stimulation (Fig. 6c) in naïve (without CPN ligation) GAD2-ires-Cre mice transfected with GtACR1-eYFP”.

Reviewer #3 (Remarks to the Author):

GENERAL COMMENTS

The paper described an impressive body of work, supporting the conclusion that the lateral parabrachial nucleus (LPB) is involved in chronic pain. Reducing enthusiasm for this manuscript are several issues with experimental design and data interpretation, and deficits in the scientific premise, reflecting a failure to adequately consider relevant, existing knowledge on the role of LPB in pain, and a

failure to relate the current findings to this knowledge.

We thank the reviewer for his/her thorough review of the manuscript. We have carefully reviewed the literature and included the missing references, including recent papers published during the submission/review.

Most of the data describe behavioral experiments that largely replicate previous findings, a worthwhile goal. Other data describe immunohistological, calcium imaging and electrophysiological experiments, some of which raise issues.

We respectfully disagree with this general comment, although we would like to reply to any specific criticism as to which experiments ‘largely replicate previous findings’. Although the LPB has been extensively investigated for its role in pain regulation, there are several novel aspects of this study:

- For the first time we demonstrated that neuropathic pain selectively activates glutamatergic but not GABAergic neurons in the LPBN, as evidenced by the CPN ligation-induced c-Fos expression and Ca²⁺ responses in glutamatergic, but not in GABAergic neurons (**Fig. 1 and Supplemental Fig. 2**).
- For the first time we systemically investigated the role of glutamatergic LPBN neurons in neuropathic pain, combined with multiple approaches: we found that optogenetic activation of glutamatergic LPBN neurons transiently induced allodynia, hyperalgesia, and place avoidance behavior in naïve mice, whereas optogenetic inhibition of these neurons transiently alleviated neuropathic pain in CPN-ligated mice, suggesting that glutamatergic LPBN neurons are both necessary and sufficient for the expression of neuropathic pain (**Figs 2–4, Supplemental Fig. 3**).
- Using virus trans-synaptic tracing and electrophysiological approaches we identified, for the first time, the direct synaptic innervation of glutamatergic

LPBN neurons by local GABAergic neurons. We further demonstrated that the activation of GABAergic LPBN neurons functionally inhibited the activity of glutamatergic LPBN neurons. In addition, we found that although CPN ligation did not affect the density of GABAergic LPBN neurons, it reduced the probability of synaptic release from these neurons (**Fig. 6i–u, Supplemental Fig. 6**).

- For the first time we demonstrate that GABAergic LPBN neurons selectively gate neuropathic pain and leave the transmission of physiological pain unaffected. Thus, optogenetic activation of GABAergic LPBN neurons did not affect physiological pain in naïve mice, but transiently induced analgesia and place preference behavior in CPN ligation-induced neuropathic pain. This was in sharp contrast with the optogenetic inhibition of glutamatergic LPBN neurons, which alleviated both physiological and neuropathic pain. Furthermore, optogenetic inactivation of GABAergic LPBN neurons transiently induced neuropathic pain symptoms in naïve mice, including allodynia, hyperalgesia, and place avoidance behavior (**Fig. 5, Fig. 6a–j, Supplemental Figs 4 and 5**)
- For the first time we showed that repetitive pharmacogenetic activation of glutamatergic LPBN neurons for one week in naïve mice induced typical chronic pain symptoms that were sustained for more than one month, whereas repetitive pharmacogenetic activation of GABAergic LPBN neurons for one week after CPN ligation prevented the development of neuropathic pain, demonstrating the critical role of glutamatergic and GABAergic LPBN neurons in the development and maintenance of neuropathic pain (**Figs 7 and 8**). Our results suggest that the LPBN is a potential key target for neuropathic pain therapy.

SPECIFIC COMMENTS

INTRODUCTION

The introduction fails to reflect current knowledge of the pathophysiology of neuropathic pain, and some of it has little to do with the topic of this study. At

places it reads like a series of unrelated statements about the pain literature, statements that do not relate to the overarching hypothesis or goal of this study. What relevant discussion about LPB circuits exists fails to reflect the rich literature on this structure and its role in pain. It is simply wrong to state that “it is not clear whether and how PBN” is involved in neuropathic pain. See, for example, PMID 8985905, 28833700, 29703377, 29862375, 31479066, 32217613.

We thank the reviewer for the critical comments and for the references listed. We carefully read the six papers the reviewer provided and note that PMID 8985905 reported changed responses of PBN neurons in the arthritic rat (inflammatory pain); PMID 28833700 reported that endogenous CGRP (strongly expressed in PBN neurons) contributes to inflammation-induced synaptic plasticity in the amygdala; PMID 29703377 is a review that does not mention neuropathic pain; PMID 29862375 reported amplified PBN neuronal activity in trigeminal neuropathic pain; whereas PMID 31479066 and 32217613 were just published during our manuscript submission and we were unaware of these two pieces of work at that time.

Our statement that “...it is not clear whether and how PBN is involved in the development and maintenance of neuropathic pain...” emphasizes the development and maintenance of neuropathic pain, which is not clear. To avoid the impression of overstatement, we rephrased the sentence as “However, the role of PBN in the development and maintenance of neuropathic pain, a persistent pain characterized by prominent emotional responses, is not clear”. We also polished some sentences in the introduction and cited more related references in the revised manuscript.

RESULTS

Figure 1

Here, and elsewhere, the authors use “noxious pinch” to activate LPB neurons,

and to compare activation patterns in animals with chronic pain with those in controls. For these analyses to be valid, a noxious stimulus that produces reproducible responses must be used. There is no indication that this was done.

We agree that a noxious stimulus inducing reproducible responses should be used for analyzing the correlation between neuronal activity and pain. We have described in more details the quantified pinch stimulation in the methods of the revised manuscript. Actually, we used pinch stimulation only in the Ca²⁺ imaging experiments. For the rest of the experiments we used the Von Frey and Hargreaves tests to analyze pain sensation and pain-associated neuronal activity. The reason that we used pinch stimulation here was that, due to technical limitations, it was difficult to induce evident Ca²⁺ activity with Von Frey mechanical stimulation or Hargreaves thermal stimulation in field Ca²⁺ imaging with fiber photometry, whereas a strong stimulus such as a pinch induced reproducible Ca²⁺ responses. To further address the reviewer's concern, we collaborated with a colleague and used a head-attached microscope for Ca²⁺ imaging with greater sensitivity so that Von Frey mechanical stimulation could be used to induce detectable Ca²⁺ activity changes in LPBN neurons. We found that CPN ligation enhanced the activity of glutamatergic LPBN neurons induced by either noxious pinch or Von Frey mechanical stimuli. These results are summarized as **Supplemental Fig. 2** in the revised manuscript.

The authors argue that CPN injury model results in specific activation of CamKIIa neurons. Yet, they do not appear to explicitly test this assertion. It seems that they need to directly compare, between sham and CPN conditions, the proportion of CamKIIa and GAD67 expressing cFos. The analyses reported here do not support the authors' conclusion.

We thank the reviewer for the constructive suggestion. Following this suggestion, we calculated the density and ratio of c-Fos-positive cells among CamKIIa and GAD67 neurons in the sham and CPN-ligated groups. We found that the ratios were very low

in the sham group (5.8% in CamKIIa- and 6.3% in GAD67-positive neurons) and no difference was found between the two types of neurons. This is in sharp contrast with the ratios of c-Fos-expressing cells in the two type neurons after CPN ligation (41.8% in CamKIIa- and 8.5% in GAD67-positive neurons). Thus, CPN ligation mainly increased c-Fos expression in CamKIIa-positive LPBN neurons. We incorporated these data into **Fig. 1c** and **d** in the revised manuscript.

Here, and in other immunohistochemical (and RNA scope) experiments, the absence of controls (e.g., omission or adsorption of primary antibody) reduces the rigor of these experiments. Also reducing rigor is the absence of information on sampling procedures, the number of sections counted, and quantification procedures.

The specificity of RNA scope staining was confirmed by immunostaining Ppib (the peptidylpropyl isomerase B encoding for cyclophilin B protein) as a positive control and dapB (dihydrodipicolinate reductase gene) as a negative control (Reference 1, see also **Fig. R2a**). We used two types of secondary antibodies (anti-rabbit Alexa Fluor 488 and anti-Guinea pig Cy3) for immunohistochemical experiments. The possibility of non-specific staining the either secondary antibody was excluded by experiments omitting the primary antibody (**Fig. R2b** and **c**). We also routinely used other approaches, such as western blots and co-immunostaining with GFP-linked target proteins in transgene mice to confirm the specificity of the primary antibodies.

We have provided more detailed information on sampling procedures, section numbers, and quantification procedures in the revised manuscript.

Fig. R2.

(a) Representative image of expression of the positive control probe (upper panel, PpiB) and the negative control probe (lower panel, dapB) in the LPBN of a wild-type mouse; scale bar, 100 μ m.

(b) Representative image of the primary antibody anti-CaMKII α (rabbit) co-immunostained with the secondary antibody anti-rabbit Alexa Fluor 488 (upper panel) and its negative control incubated only with anti-rabbit Alexa Fluor 488 (lower panel) in the LPBN of a wild-type mouse; scale bar, 100 μ m.

(c) Representative image of the primary antibody anti-c-Fos (Guinea pig) co-immunostained with the secondary antibody anti-Guinea pig Cy3 (upper panel) and only incubated with anti-Guinea pig Cy3 (lower panel) in the LPBN of Vglut2-ires-Cre mice injected with the AAV-EF1a-DIO-eYFP virus; scale bar, 100 μ m.

Contrary to the stated assumption, DAPI is not a neuronal marker, and should not be used as a proxy for one.

We thank the reviewer for pointing this out. In most of our experiments, we used DAPI staining to determine cell-associated fluorescence signals so that non-specific signals could be excluded. In the revised manuscript we calculated the percent of GABAergic neurons as the ratio of GAD1-positive cells / GAD1-positive cells + Vglut2-positive cells in the LPBN.

It is inaccurate to state that “glutamatergic LPB neurons are selectively activated in the mechanical allodynia”. What the authors show is that glutamatergic neurons in LPB are more active in CPN injured animals in response to a single hindpaw pinch stimulus. This is not allodynia.

We rephrased the sentence as “glutamatergic LPBN neurons are selectively activated by the noxious stimulus”.

Figure 2

Feasibility experiments for optogenetic stimulation show that opsin-containing neurons can entrain stimuli at 5Hz stimuli. Behavioral experiments use stimuli of 20Hz, without ascertaining that LPB neurons can entrain such stimuli.

We conducted new experiments and confirmed that LPBN neurons respond nicely to 20-Hz light stimulation (**Fig. 2b** in the revised manuscript).

The finding that tests of anxiety were not affected by the induction of chronic pain are surprising, and contradict literature indicating otherwise. This should be considered and discussed.

We agree that fully-developed chronic pain may be accompanied by anxiety. Although our 5-min optogenetic stimulation instantly induced pain behavior, this was not

fully-developed chronic pain. We have briefly discussed this issue in the revised manuscript.

The authors state that optogenetically activating glutamatergic neurons causes hyperalgesia; that is inaccurate. They present evidence for allodynia and aversiveness, but not for hyperalgesia.

Because we found that optogenetically activating glutamatergic neurons not only reduced the threshold of the paw withdraw response in the Von Frey test (mechanical allodynia, **Fig. 2d**), but also reduced the latency of the paw withdrawal response in the Hargreaves test (heat hyperalgesia, **Fig. 2e**) We used the term “hyperalgesia” for general description in the original manuscript. In the revised manuscript, we describe these results separately using more precise terms.

The statement that activating glutamatergic neurons causes signs of pain is not “striking”, as the authors state. There exists substantial prior research indicating that activating excitatory neurons in LPB causes pain-like behaviors, as well as pronounced aversion.

We rephrased the sentence in the revised manuscript.

Figure 4 (Figure 5 in the revised manuscript)

There is no explicit consideration of the likelihood that optogenetic stimulation resulted in antidromic activation of neurons, and their axon collaterals, in other nuclei. This is a potential confound to both the experiments using the vGAT-hChR2 mice and those using the GAD2-Cre mouse with local injection of AAV-hChR2s. Therefore, it is wrong to assume that optical stimuli exclusively activate neurons intrinsic to LPB. Similarly, the assumption that local infusion of picrotoxin selectively inhibits somata needs to be tempered by knowledge that picrotoxin is also

a channel blocker, and can affect activity of afferents extrinsic to LPB.

We agree that optogenetic stimulation may activate axons of passage originating from other nuclei when the ChR2 transgenic mouse is used. However, such a concern can be excluded when using the GAD2-Cre mouse with local injection of AAV-hChR2 virus into the LPBN, because only GABAergic neurons in the LPBN express ChR2 and are activated by light stimulation. We carefully examined the virus injection site and data were excluded if we found that virus infection was not limited to the LPBN (see supplemental Fig. 4 for depiction of the superimposed virus infection areas from six GAD2-ires-Cre mice bilaterally injected with AAV-DIO-ChR2-eYFP virus in the LPBN). The GABA_A receptor itself is a Cl⁻ channel and picrotoxin is a specific blocker of the Cl⁻ channel of the GABA_A receptor, rather than a blocker of any other type of channel. Our result that local injection of picrotoxin into the LPBN blocked the effects of light activation of GABAergic LPBN neurons also excluded the possibility that GABAergic LPBN neurons act through their projections outside the LPBN. Thus, we have used all the available approaches in the field, which to our knowledge, should be able to exclude the concerns the reviewer raised.

The paper might benefit from referencing important papers on the role of GABAergic inputs from CeA to LPB. For example, PMID 31577943, 26733798, 32217613.

We thank the reviewer for providing these references. We did point out in the original manuscript that the LPBN receives GABAergic innervation from other brain regions. We checked these three references and found that PMID 32217613 (published in April, 2020), which reported a role of GABAergic input from the CeA to the LPB in chronic pain (An Amygdalo-Parabrachial Pathway Regulates Pain Perception and Chronic Pain, J Neurosci, 2020 Apr 22;40:3424-3442), was most relevant to our study. The other two papers (PMID 31577943 and 26733798) only reported GABAergic neurons in the CeA, but did not describe their projections to the LPB. We thus cited PMID

32217613 in our revised manuscript.

Figure 5 (Figure 6 in the revised manuscript)

The stated goal of the experiments described here is to test the hypothesis that GABAergic LPB neurons “gate neuropathic pain”. The relevant experiment here would have been to compare data from sham and CPN conditions. This experiment was not performed.

The hypothesis that GABAergic LPBN neurons gate neuropathic pain is the major point of our paper. We have thus provided multiple pieces of solid data presented in several figures including Fig. 6 (Fig. 5 in the first version of the manuscript). In this Fig. 6 we show that optogenetic inactivation of GABAergic LPBN neurons induces mechanical allodynia, heat hyperalgesia, and aversive behavior, mimicking neuropathic pain. The data for the experiments ‘comparing data from sham and CPN conditions’ are presented in several other figures (e.g. Figs 5 and 7) in the manuscript. Actually, the major data presented in Fig. 6 are results showing ‘GABAergic LPBN neurons monosynaptically innervate local glutamatergic neurons’. We have modified the title of Fig. 6 in the revised manuscript.

The comparisons of labeled neurons is also not consistent with the relevant statistical hypothesis. A meaningful comparison here would be to compare the percentage of c-fos positive GAD2 neurons in GtACR1 versus EYFP, and to separately compare the percentage of c-Fos positive CamkIIa neurons in GtACR1 versus EYFP.

We thank the reviewer for this suggestion. In the revised manuscript we also present data in **Fig. 6h** comparing the percentage of c-fos positive GAD2 and CamkIIa neurons between GtACR1 and EYFP mice.

In patch-clamp experiments it is surprising that large IPSCs were evoked while holding neurons near the chloride reversal potential. It might be instructive to discuss this finding.

We used high Cl^- (135 mM) pipette solution for whole-cell patch recordings and IPSCs were thus recorded as an inward current when the membrane potential was held at -70 mV, which was much more negative than the Cl^- reversal potential under our conditions. We described the recording conditions in the methods in the original manuscript. We also briefly discuss this issue in the results of the revised manuscript.

Data analysis

- *There is no justification for the use of parametric statistics.*

Since our data were continuous variables, parametric statistics was first considered. Before ANOVA analysis was performed, the homogeneity of variance was examined by Levene's test of Equality of Error Variances. One- or two-way ANOVA was used for data that met the homogeneity of variance test. A square root transformation or logarithmic transformation was performed for data that did not meet the homogeneity of variance test. Two-way ANOVA was then performed if data met the homogeneity of variance test after transformation. The Levene's Statistic and the significance values of all the data, including those after transformation, are listed in Table R1 and $P > 0.05$ indicates a significant difference. We found $p > 0.05$ in all of our data. For details, please see Table R1 below. We have described this issue briefly in the methods of the revised manuscript.

Table R1

Levene' s test of Equality of Error Variances					
	Levene Statistic	df1	df2	Sig.(significance)	
Figure 1c	Based on median and with adjusted df	3.871	3	4.740	0.095
Figure 1d	Based on median and with adjusted df	1.979	3	6.662	0.21
Figure 1e	Based on median and with adjusted df	2.675	3	10.548	0.101
Figure 1k left	Based on median and with adjusted df	1.448	3	4.472	0.344
Figure 1k right	Based on median and with adjusted df	0.177	3	14.895	0.91
Figure 2d sham	Based on median and with adjusted df	0.638	3	33.213	0.596
Figure 2d CPN	Based on median and with adjusted df	1.16	3	33.838	0.339
Figure 2e	Based on median and with adjusted df	1.255	7	43.861	0.295
Figure 2h	Based on median and with adjusted df	0.347	7	10.599	0.714
Figure 2j	Based on median and with adjusted df	2.466	2	12.14	0.126
Figure 2l	Based on median and with adjusted df	1.197	5	31.8	0.333

Figure 2m

	Levene Statistic	df1	df2	Sig.(significance)
Based on median and with adjusted df	0.741	5	43.982	0.597

Figure 2n

	Levene Statistic	df1	df2	Sig.(significance)
Based on median and with adjusted df	0.296	2	13.713	0.748

Figure 2o

	Levene Statistic	df1	df2	Sig.(significance)
Based on median and with adjusted df	2.644	2	11.971	0.112

Figure 2r upper

	Levene Statistic	df1	df2	Sig.(significance)
Based on median and with adjusted df	0.551	5	12.553	0.735

Figure 2r lower

	Levene Statistic	df1	df2	Sig.(significance)
Based on median and with adjusted df	0.796	5	15.4	0.569

Figure 3b
Square root
transformation

	Levene Statistic	df1	df2	Sig.(significance)
Based on median and with adjusted df	1.756	7	26.198	0.139

Figure 3c

	Levene Statistic	df1	df2	Sig.(significance)
Based on median and with adjusted df	0.904	7	55.59	0.51

Figure 3f left

	Levene Statistic	df1	df2	Sig.(significance)
Based on median and with adjusted df	1.215	2	8.81	0.342

Figure 3f right

	Levene Statistic	df1	df2	Sig.(significance)
Based on median and with adjusted df	2.447	2	7.031	0.156

Figure 3i upper

	Levene Statistic	df1	df2	Sig.(significance)
Based on median and with adjusted df	1.074	2	25.76	0.357

Figure 3i lower

	Levene Statistic	df1	df2	Sig.(significance)
Based on median and with adjusted df	0.03	2	25.245	0.97

Figure 3j upper

	Levene Statistic	df1	df2	Sig.(significance)
Based on median and with adjusted df	0.516	5	10.186	0.759

Figure 3j lower

	Levene Statistic	df1	df2	Sig.(significance)
Based on median and with adjusted df	0.859	5	10.259	0.539

Figure 4b
Square root transformation

	Levene Statistic	df1	df2	Sig.(significance)
Based on median and with adjusted df	1.147	7	58.891	0.347

Figure 4e left

	Levene Statistic	df1	df2	Sig.(significance)
Based on median and with adjusted df	1.754	2	10.744	0.219

Figure 4e right

	Levene Statistic	df1	df2	Sig.(significance)
Based on median and with adjusted df	0.099	2	10.492	0.907

Figure 4g upper

	Levene Statistic	df1	df2	Sig.(significance)
Based on median and with adjusted df	0.874	5	17.515	0.518

Figure 4g lower

	Levene Statistic	df1	df2	Sig.(significance)
Based on median and with adjusted df	1.827	5	13.196	0.176

Figure 4h upper

	Levene Statistic	df1	df2	Sig.(significance)
Based on median and with adjusted df	2.031	5	27.4	0.106

Figure 4h lower

	Levene Statistic	df1	df2	Sig.(significance)
Based on median and with adjusted df	1.695	5	20.528	0.181

Figure 5b
Square root transformation

	Levene Statistic	df1	df2	Sig.(significance)
Based on median and with adjusted df	1.043	5	29.747	0.411

Figure 5e

	Levene Statistic	df1	df2	Sig.(significance)
Based on median and with adjusted df	2.368	2	8.637	0.949

Figure 5g

	Levene Statistic	df1	df2	Sig.(significance)
Based on median and with adjusted df	3.247	2	10.838	0.079

Figure 5i

	Levene Statistic	df1	df2	Sig.(significance)
--	------------------	-----	-----	--------------------

logarithmic transformation	Based on median and with adjusted df	1.504	7	24.467	0.212
--------------------------------------	-------	---	--------	-------

Figure 5j		Levene Statistic	df1	df2	Sig.(significance)
	Based on median and with adjusted df	2.368	7	8.362	0.121

Figure 5k left		Levene Statistic	df1	df2	Sig.(significance)
	Based on median and with adjusted df	0.427	2	13.782	0.661

Figure 5k right		Levene Statistic	df1	df2	Sig.(significance)
	Based on median and with adjusted df	1.765	2	13.307	0.209

Figure 5l upper		Levene Statistic	df1	df2	Sig.(significance)
	Based on median and with adjusted df	0.581	5	21.534	0.714

Figure 5l lower		Levene Statistic	df1	df2	Sig.(significance)
	Based on median and with adjusted df	0.254	5	17.235	0.932

Figure 5o		Levene Statistic	df1	df2	Sig.(significance)
	Based on median and with adjusted df	1.205	20	100.71	0.267

Figure 6b logarithmic transformation		Levene Statistic	df1	df2	Sig.(significance)
	Based on median and with adjusted df	1.887	5	30.036	0.153

Figure 6c logarithmic transformation		Levene Statistic	df1	df2	Sig.(significance)
	Based on median and with adjusted df	2.257	5	30.327	0.102

Figure 6f left		Levene Statistic	df1	df2	Sig.(significance)
	Based on median and with adjusted df	0.077	5	20.618	0.995

Figure 6f right		Levene Statistic	df1	df2	Sig.(significance)
	Based on median and with adjusted df	0.729	5	15.098	0.612

Figure 7b		Levene Statistic	df1	df2	Sig.(significance)
	Based on median and with adjusted df	1.983	40	65.036	0.065

Figure 7g

	Levene Statistic	df1	df2	Sig.(significance)
Based on median and with adjusted df	1.398	47	95.012	0.085

Figure 7i

	Levene Statistic	df1	df2	Sig.(significance)
Based on median and with adjusted df	1.455	39	60.129	0.094

Figure 7k

	Levene Statistic	df1	df2	Sig.(significance)
Based on median and with adjusted df	0.68	3	6.163	0.595

- *No a priori sample size estimation was performed. Stating that “sample sizes are similar to those reported in previous publications” does not reduce this deficiency.*

The sample sizes were determined by common practice in the field. Nevertheless, the statistical significance of the results we obtained justifies the sample size we used.

- *Several important figures fail to depict individual data points.*

Following the reviewer’s suggestion, we present individual data points in all the figures.

- *Both Bonferroni and Tukey post-hoc corrections are used; why?*

We thank the reviewer for raising this question. We now use Bonferroni’s test for unified statistical analysis of all the data in the revised manuscript. Although the P values of the data analyzed with Bonferroni’s test are slightly different from those analyzed with Tukey test, the conclusions of statistical significance are same using either test. For convenience of comparison, the P values of the data previously analyzed with Tukey’s test (previous version of the manuscript) are listed together with that analyzed with Bonferroni’s test (revised manuscript) in Table R2 below.

Table R2. Comparison of P values of the data analyzed with Tukey's (black) and Bonferroni's test (blue)

Figure 1 k, n

Two-way repeated measures ANOVA followed by **Tukey's or Bonferroni's** multiple comparisons test:

(**k**, left) Sham vs CPN (eYFP), $P = 0.9919$ (>0.9999); GCaMP7s, $**P = 0.0046$ (<0.0001); eYFP vs GCaMP7s (Sham), $*P = 0.0176$ (**0.0109**); CPN, $***P = 0.0004$ (<0.0001). (**k**, right), Sham vs CPN (eYFP), $P = 0.6754$ (**0.3970**); GCaMP7s, $**P = 0.0036$ (<0.0001); eYFP vs GCaMP7s (Sham), $*P = 0.0295$ (**0.0102**); CPN, $***P = 0.0004$ (<0.0001). (**n**, left) Sham vs CPN (eYFP), $P = 0.8417$ (>0.9999); GCaMP7s, $P = 0.9730$ (>0.9999); eYFP vs GCaMP7s (Sham), $*P = 0.0189$ (**0.0197**); CPN, $P = 0.5684$ (**0.0610**). (**n**, right), Sham vs CPN (eYFP), $P = 0.6754$ (>0.9999); GCaMP7s, $P = 0.0836$ (**0.5367**); eYFP vs GCaMP7s (Sham), $*P = 0.0295$ (**0.0234**); CPN, $P = 0.0547$ (**0.2663**);

Figure 2 h j

One-way repeated measures ANOVA followed by Tukey's **or Bonferroni's** multiple comparisons test. (**h**) Pre vs Light, $P = 0.9234$ (>0.9999); Pre vs Post, $P = 0.1105$ (**0.4438**); Light vs Post, $P = 0.7043$ (**0.9724**); (**j**) Pre vs Light, $***P = 0.0004$ (<0.0001); Pre vs Post, $**P = 0.0024$ (**0.0006**); Light vs Post, $**P = 0.0057$ (**0.0002**)

Figure 3f

One-way repeated measures ANOVA followed by Tukey's **or Bonferroni's** multiple comparisons test: Left, (mCherry), Pre vs Light, $P = 0.6621$ (**0.9418**); Pre vs Post, $P = 0.9958$ (>0.9999); Light vs Post, $P = 0.5750$ (>0.9999); Right (NpHR), Pre vs Light, $**P = 0.0041$ (**0.0047**); Pre vs Post, $*P = 0.0267$ (**0.0319**); Light vs Post, $P = 0.5656$ (**0.9531**);

Figure 3i, j

Two-way ANOVA followed by **Tukey's or Bonferroni's** multiple comparisons test in (i): upper, mCherry vs eNpHR (Pre, $P = 0.1279$ (0.1055); Light, $P = 0.4076$ (0.4452); Post, $P = 0.2540$ (0.2445); mCherry (Pre vs Light), $P = 0.8710$ (> 0.9999); (Pre vs Post), $P = 0.4983$ (0.7484); (Light vs Post), $P = 0.7963$ (> 0.9999); eNpHR (Pre vs Light), $P = 0.9374$ (> 0.9999); (Pre vs Post), $P = 0.7844$ (> 0.9999); (Light vs Post), $P = 0.5801$ (0.1895); Lower, mCherry vs eNpHR (Pre, $P = 0.5178$ (> 0.9999); Light, $P = 0.7642$ (> 0.9999); Post, $P = 0.5459$ (0.7001)); mCherry (Pre vs Light), $P = 0.9502$ (0.6961); (Pre vs Post), $P = 0.3657$ (> 0.9999); (Light vs Post), $P = 0.5324$ (0.9471); eNpHR (Pre vs Light), $P = 0.9925$ (> 0.9999); (Pre vs Post), $P = 0.4381$ (0.7485); (Light vs Post), $P = 0.5038$ (0.8849). (j): upper, mCherry vs eNpHR (Pre, $P = 0.5520$ (0.6075); Light, $P = 0.9983$ (> 0.9999); Post, $P = 0.5827$ (0.8437)); mCherry (Pre vs Light), $P = 0.1511$ (0.0520); (Pre vs Post), $P = 0.5103$ (0.5881); (Light vs Post), $P = 0.6648$ (0.6963); eNpHR (Pre vs Light), $P = 0.6179$ (> 0.9999); (Pre vs Post), $P = 0.5392$ (0.8645); (Light vs Post), $P = 0.9903$ (> 0.9999). Lower, mCherry vs eNpHR (Pre, $P = 0.0656$ (0.1582); Light, $P = 0.6819$ (> 0.9999); Post, $P = 0.7142$ (> 0.9999)); mCherry (Pre vs Light), $P = 0.8108$ (> 0.9999); (Pre vs Post), $P = 0.9593$ (> 0.9999); (Light vs Post), $P = 0.9353$ (> 0.9999); eNpHR (Pre vs Light), $P = 0.7908$ (> 0.9999); (Pre vs Post), $P = 0.5642$ (0.7637); (Light vs Post), $P = 0.9183$ (> 0.9999).

Figure 4c (Figure 5c in the revised manuscript)

Two-way repeated measures ANOVA followed by **Tukey's or Bonferroni's** multiple comparisons test: Sham (Pre vs Light), $P = 0.9930$ (>0.9999); (Light vs Post), $P = 0.5715$ (>0.9999); CPN (Pre vs Light), $**P = 0.0021$ (<0.0001); (Light vs Post), $**P = 0.0052$ (<0.0001); Sham vs CPN (Pre, $***P = 0.0009$ (<0.0001); Light, $P > 0.9999$ (0.1903), Post, $*P = 0.0230$ (<0.0001),

Figure 4l (Figure 5l in the revised manuscript)

Two-way ANOVA followed by **Tukey's or Bonferroni's** multiple comparisons test in (upper): eYFP vs Chr2 (Pre, $P = 0.996$ (0.2524); Light, $P = 0.8218$ (0.1670); Post, $P = 0.9789$ (0.0833); eYFP (Pre vs Light), $P = 0.4426$ (>0.9999); (Pre vs Post), $P =$

0.9992 (0.4933); (Light vs Post), $P = 0.4222$ (>0.9999); ChR2 (Pre vs Light), $P = 0.999$ (>0.9999); (Pre vs Post), $P = 0.9852$ (>0.9999); (Light vs Post), $P = 0.9768$ (>0.9999). Lower, eYFP vs ChR2 (Pre, $P = 0.8434$ (>0.9999); Light, $P = 0.6636$ (>0.9999); Post, $P = 0.7856$ (>0.9999); eYFP (Pre vs Light), $P = 0.6534$ (>0.9999); (Pre vs Post), $P = 0.8017$ (>0.9999); (Light vs Post), $P = 0.9321$ (>0.9999); ChR2 (Pre vs Light), $P = 0.8763$ (>0.9999); (Pre vs Post), $P = 0.9534$ (>0.9999); (Light vs Post), $P = 0.7423$ (>0.9999);

Figure 5f (Figure 6f in the revised manuscript)

Two-way ANOVA followed by **Tukey's or Bonferroni's** multiple comparisons test in (left): DIO-eYFP vs DIO-GtACR1 (Pre, $P = 0.1279$ (0.0610); Light, $P = 0.4076$ (0.3569); Post, $P = 0.2540$ (0.1740); DIO-eYFP (Pre vs Light), $P = 0.8710$ (>0.9999); (Pre vs Post), $P = 0.4983$ (0.7484); (Light vs Post), $P = 0.7963$ (>0.9999); DIO-GtACR1 (Pre vs Light), $P = 0.9374$ (>0.9999); (Pre vs Post), $P = 0.7844$ (>0.9999); (Light vs Post), $P = 0.5801$ (0.9286); (right), DIO-eYFP vs DIO-GtACR1 (Pre, $P = 0.5178$ (0.7191); Light, $P = 0.7642$ (>0.9999); Post, $P = 0.5459$ (0.7700); $F(2, 8) = 2.011$; DIO-eYFP (Pre vs Light), $P = 0.9502$ (>0.9999); (Pre vs Post), $P = 0.3657$ (0.6744); (Light vs Post), $P = 0.5324$ (>0.9999); DIO-GtACR1 (Pre vs Light), $P = 0.9925$ (>0.9999); (Pre vs Post), $P = 0.4381$ (0.8193); (Light vs Post), $P = 0.5038$ (0.9571);

Figure 6i (Figure 7i in the revised manuscript)

Two-way repeated measures ANOVA followed by **Tukey's or Bonferroni's** multiple comparisons test: DIO-mCherry CPN vs DIO-hM3Dq-mCherry CPN, Baseline, $P = 0.598$ (>0.9999); 1, $P = 0.9811$ (>0.9999); 3, $*P = 0.0189$ (0.0063); 5, $**P = 0.0021$ (0.0013); 7, $***P = 0.0002$ (<0.0001); 9, $****P < 0.0001$ (<0.0001); 11, $****P < 0.0001$ (<0.0001); 13, $****P < 0.0001$ (<0.0001); 15, $****P < 0.0001$ (<0.0001); 28, $****P < 0.0001$ (<0.0001); DIO-hM3Dq-mCherry Sham (blue line) vs DIO-hM3Dq-mCherry CPN (red line), Baseline, $P = 0.9295$ (>0.9999); 1, $P = 0.8081$ (>0.9999); 3, $P = 0.9920$ (>0.9999); 5, $P = 0.7477$ (>0.9999); 7, $P = 0.9997$ (>0.9999);

9, $P = 0.8128$ (>0.9999); 11, $P = 0.9473$ (>0.9999); 13, $P = 0.9970$ (>0.9999); 15, $P = 0.9826$ (>0.9999); 28, $P = 0.9959$ (>0.9999);

Figure 6k (Figure 7k in the revised manuscript)

One-way repeated measures ANOVA followed by **Tukey's or Bonferroni's** multiple comparisons test: mCherry Sham vs mCherry CPN, **** $P < 0.0001$ (<0.0001); mCherry Sham vs hM3Dq Sham, $P = 0.447$ (0.9235); mCherry Sham vs hM3Dq CPN, $P = 0.5647$ (0.4521); mCherry CPN vs hM3Dq Sham, *** $P = 0.0001$ (<0.0001); mCherry CPN vs hM3Dq CPN, *** $P = 0.0001$ (<0.0001); hM3Dq Sham vs hM3Dq CPN, $P = 0.9957$ (>0.9999).

DISCUSSION

It would be appropriate to revise the discussion after the analyses suggested above are implemented.

We have revised our manuscript, including the discussion.

It would be useful and appropriate to consider the fact that LPB interacts with a large number of pain-related brain regions. This should temper the strong conclusion that manipulating PB neurons is both necessary and sufficient to induce neuropathic pain.

We agree that many pain-related brain regions interact, just like any other brain function-associated regions, including those that are both necessary and sufficient for specific brain functions. We have polished the discussion of the revised manuscript.

Reference

1. Fay Wang et al, RNAscope: a novel in situ RNA analysis platform for formalin-fixed, paraffin-embedded tissues, J Mol Diagn, 2012,14: 22-9.

REVIEWERS' COMMENTS

Reviewer #1 (Remarks to the Author):

The authors are very responsive and have conducted many new experiments. Many panels of new data are included in the revision. The authors used a head-attached microscope for in vivo Ca²⁺ imaging and demonstrated mechanical stimulation-induced Ca²⁺ signaling in glutamatergic LPBN neurons (Supplemental Fig. 2). Importantly, the new data showed that CPN ligation did not change the density of GABAergic LPBN neurons but decreased the release probability of these neurons as evidenced by an increased paired-pulse ratio of evoked IPSCs, suggesting a possible disinhibition of LPBN neurons in neuropathic pain (Supplemental Fig. 6). Furthermore, they showed that pharmacogenetic activation of GABAergic LPBN neurons in CPN-ligated mice also induced conditioned place-preference behavior, (Supplementary Fig. 5). The potential limitations of this study are also discussed. The authors also included a table for statistical analyses in the response letter. Overall, these new data consolidate the major conclusions of this study. LPBN neurocircuit is a hot topic of recent pain research, but this study is comprehensive and has provided new insights into excitatory and inhibitory circuit modulation of neuropathic pain in the LPBN.

Reviewer #2 (Remarks to the Author):

The authors have satisfyingly implicated all comments and suggestions. I have no further questions.

Reviewer #3 (Remarks to the Author):

The authors have responded adequately to previous critiques.